# Interpretation of Measured Aerosol Mass Scattering Efficiency Over North America Using a Chemical Transport Model

Robyn N. C. Latimer[1*], Randall V. Martin[1,2*]

[1]Department of Physics and Atmospheric Science, Dalhousie University, Halifax, B3H 4R2
[2]Harvard-Smithsonian Center for Astrophysics, Cambridge, MA 02138, USA

*Correspondence to*: Robyn Latimer (robyn.latimer93@gmail.ca) and Randall Martin (randall.martin@dal.ca)

**Abstract.** Aerosol mass scattering efficiency affects climate forcing calculations, atmospheric visibility, and the interpretation of satellite observations of aerosol optical depth. We evaluated the representation of aerosol mass scattering efficiency ($\alpha_{sp}$) in the GEOS-Chem chemical transport model over North America using collocated measurements of aerosol scatter and mass from IMPROVE network sites between 2000-2010. We found a positive bias in mass scattering efficiency given current assumptions of aerosol size distributions and particle hygroscopicity in the model. We found that overestimation of mass scattering efficiency was most significant in dry (RH<35%) and midrange humidity (35%<RH<65%) conditions, with biases of 82% and 40%, respectively. To address these biases, we investigated assumptions surrounding the two largest contributors to fine aerosol mass, organic (OA) and secondary inorganic aerosols (SIA). Inhibiting hygroscopic growth of SIA below 35% RH and decreasing the dry geometric mean radius, from 0.069 μm for SIA and 0.073 μm for OA to 0.058 μm for both aerosol types, significantly decreased the overall bias observed at IMPROVE sites in dry conditions from 82% to 9%. Implementation of a widely used alternative representation of hygroscopic growth following κ-Kohler theory for secondary inorganic (hygroscopicity parameter κ=0.61) and organic (κ=0.10) aerosols eliminated the remaining overall bias in $\alpha_{sp}$. Incorporating these changes in aerosol size and hygroscopicity into the GEOS-Chem model resulted in an increase of 16% in simulated annual average $\alpha_{sp}$ over North America, with larger increases of 25% to 45% in northern regions with high RH and hygroscopic aerosol fractions, and decreases in $\alpha_{sp}$ up to 15% in the southwestern U.S. where RH is low.

## 1. Introduction

The interaction of atmospheric aerosols with radiation has substantial implications for the direct radiative effects of atmospheric aerosols, atmospheric visibility, and satellite retrievals of aerosol optical properties. The direct radiative effects of aerosols remain a major source of uncertainty in radiative forcing (Myhre et al., 2013). Atmospheric visibility affects the appearance of landscape features, which is of particular concern in national parks and wilderness areas (Malm et al., 1994). Gaining insight into the concentration and composition of atmospheric aerosols via interpretation of satellite retrievals of

aerosol optical depth (AOD) also relies heavily on an understanding of the interaction of aerosols with radiation (Kahn et al., 2005). Analysis of collocated measurements of aerosol scatter, mass, and composition could offer valuable insight into aerosol optical properties.

Mass scattering efficiency is a complex function of aerosol size, composition, hygroscopicity and mixing state (Hand and Malm, 2007; Malm and Kreidenweis, 1997; White, 1986). Current chemical transport models and global circulation models often calculate atmospheric extinction due to aerosols from speciated aerosol mass concentrations using a composition and size dependent mass extinction efficiency ($\alpha_{ext}$, $m^2$ $g^{-1}$). Many of these models use aerosol optical and physical properties defined by the Global Aerosol Data Set (GADS), compiled from measurements and models from 1970 to 1995 (Koepke et al., 1997). The subsequent expansion in long term aerosol monitoring offers an exciting possibility to further improve model representation of aerosol physical and optical properties. The Interagency Monitoring of Protected Visual Environments (IMPROVE) network offers long-term collocated measurements since 1987 of particle scatter ($b_{sp}$), relative humidity (RH), particulate mass concentrations less than 10 µm ($PM_{10}$) and less than 2.5 µm ($PM_{2.5}$) as well as $PM_{2.5}$ chemical composition at sites across the United States and Canada (Malm et al., 1994; Malm et al., 2004). These collocated measurements provide direct estimates of mass scattering efficiency ($\alpha_{sp}$) across North America that are useful to evaluate and improve the mass scattering efficiency currently used in models.

Several prior studies have analysed mass scattering efficiencies. Hand et al. (2007) performed an extensive review that examined and compared mass scattering efficiencies calculated from ground based measurements from approximately 60 mostly short-term studies from 1990 to 2007. In this review, the importance of long term measurements was emphasized. Malm & Hand (2007) applied IMPROVE network data between 1987-2003 to evaluate mass scattering efficiency of organic and inorganic aerosols at 21 IMPROVE sites. A couple more recent examples of short term studies of mass scattering efficiency are Titos et al. (2012) and Tao et al. (2014). Many other long-term multi-site studies have investigated aerosol optical properties (e.g. Andrews et al., 2011; Coen et al., 2013; Pandolfi et al., 2017), but few include measurements of aerosol mass concentrations and therefore do not provide information on mass scattering efficiencies. Our study builds upon previous studies of mass scattering efficiency by reducing initial assumptions regarding size and hygroscopicity of inorganic and organic aerosols and by using measurements of particle speciation, mass and scatter to inform the representation of these properties. We interpret long term measurement data to obtain a representation of mass scattering efficiency that can be used across an array of conditions and locations to facilitate incorporation into chemical transport models.

Here we interpret collocated measurements of $PM_{2.5}$, $PM_{10}$, $b_{sp}$ and RH from the IMPROVE network to understand factors affecting the representation of mass scattering efficiency. Section 2 provides a description of IMPROVE network measurements, of the GEOS-Chem chemical transport model, and of an alternate aerosol hygroscopic growth scheme. In Section 3, we present an analysis of the current representation of mass scattering efficiency in the GEOS-Chem model, and identify changes that improve the consistency with observations. The impact of these changes on GEOS-Chem simulated mass scattering efficiency, as well as on agreement between the GEOS-Chem model and observations from the IMPROVE network are described in section 4.

## 2. Methods

### 2.1 IMPROVE network measurements

The IMPROVE network (Malm et al., 1994) is a long-term monitoring program established in 1987 to monitor visibility trends in national parks and wilderness areas in the United States. The network offers measurements of $PM_{2.5}$ speciation, $PM_{2.5}$ and $PM_{10}$ gravimetric mass, and collocated measurements of $b_{sp}$ and RH at a subset of sites that we interpret to understand mass scattering efficiency.

The IMPROVE particle sampler collects $PM_{2.5}$ and $PM_{10}$ on filters. Sampling occurs over a 24h period every third day. Collected $PM_{2.5}$ is analyzed for fine gravimetric mass, elemental concentrations (including Al, Si, Ca, Fe, Ti), ions ($SO_4^{2-}$, $NO_3^-$, $NO_2^-$, $Cl^-$), and organic and elemental carbon. Collected $PM_{10}$ undergoes gravimetric analysis for total particulate mass less than 10 μm, allowing for the determination of coarse mass ($PM_{10}$-$PM_{2.5}$) (Malm et al., 1994).

Particle scatter ($b_{sp}$) is measured at 550 nm at a subset of IMPROVE sites using OPTEC NGN-2 open air integrating nephelometers (Malm et al., 1994; Malm and Hand, 2007; Molenar, 1997). $b_{sp}$ is reported hourly at ambient air temperature and relative humidity; all three parameters are recorded. We filter $b_{sp}$ data to exclude measurements likely affected by meteorological interference such as fog. These conditions include an RH threshold of 95%, a maximum $b_{sp}$ threshold of 5000 Mm$^{-1}$ and an hourly rate of change threshold for $b_{sp}$ of 50 Mm$^{-1}$, following IMPROVE filtering protocols (IMPROVE, 2004).

The IMPROVE network collects collocated samples at a subset of sites, which can provide insight into precision errors associated with the measurements of major species. Hyslop and White (2008) and Solomon et al. (2014) found mean collocated precision errors ranging from 6-11% for particulate mass measured by IMPROVE. Typical uncertainties in IMPROVE $b_{sp}$ measurements are in the range of 5-15% (Gebhart et al., 2001). Due to nephelometer truncation errors, uncertainties in measured $b_{sp}$ increase as particle size distributions increase, and coarse particle scattering can be underestimated (Molenar, 1997).

For this study, we select sites where fine aerosol mass and speciation measurements are collocated with IMPROVE nephelometers between 2000-2010. We exclude data after 2010 to address concerns about variable laboratory RH for $PM_{10}$ measurement after 2010. Sea salt aerosols are excluded from the analysis from 2000-2004, as reliable estimates of sea salt concentrations were not reported during this period. We exclude coastal sites during this period, as sea salt can contribute significantly to $b_{sp}$ in coastal conditions of high RH due to its highly hygroscopic nature (Lowenthal and Kumar, 2006). We use only days with coincident mass and scatter measurements, and a minimum of 23 hourly measurements per day, to reduce influence of meteorological interference. Additionally, only sites with a minimum of 90 days of measurements are included in the analysis.

Figure 1 shows at the 28 sites used in this study the average hourly $b_{sp}$ at ambient RH and the average 24h $PM_{10}$ and $PM_{2.5}$ measured between 2000-2010. Measured $b_{sp}$ values vary by a factor of 7 with scatter below 20 Mm$^{-1}$ across the southwest U.S., and scatter above 50 Mm$^{-1}$ across the southeast U.S. Measured $PM_{10}$ concentrations vary by a factor of 3

with values below 6 µg m$^{-3}$ in the west to above 14 µg m$^{-3}$ in the southeast. Measured PM$_{2.5}$ concentrations also vary by a factor of 3, with values below 3 µg m$^{-3}$ in the west to above 9 µg m$^{-3}$ in the southeast.

## 2.2 GEOS-Chem simulation

We simulate hourly PM$_{2.5}$ and PM$_{10}$ mass concentrations and particle scatter using the global chemical transport model
GEOS-Chem (version 11.01, http://geos-chem.org). The GEOS-Chem model is driven by assimilated meteorology from the Goddard Earth Observation System (GEOS MERRA-2, (Gelaro et al., 2017) of the NASA Global Modeling and Assimilation Office (GMAO). Our simulation for North America is conducted at 2° x 2.5° resolution over 47 vertical levels.

The majority of our analysis focuses on the accuracy of the GEOS-Chem parameterization of mass scattering efficiency based on optical parameters given in Table A1. These default aerosol physical and optical properties are defined
by the Global Aerosol Data Set (GADS) (Koepke et al., 1997), as implemented by Martin et al. (2003), with modifications to dry size distributions (Drury et al., 2010) and dust mass partitioning (Ridley et al., 2012). After evaluating and improving this parameterization, implications are examined using the full GEOS-Chem simulation in section 3.3.

GEOS-Chem simulates detailed aerosol-oxidant chemistry (Bey et al., 2001; Park et al., 2004). The aerosol simulation includes the sulfate-nitrate-ammonium system (Park et al., 2004), primary (Park et al., 2003; Wang et al., 2014)
and secondary (Pye et al., 2010) carbonaceous aerosols, mineral dust (Fairlie et al., 2007; Fairlie et al., 2010; Zhang et al., 2013) and sea salt (Jaeglé et al., 2011). Organic matter (OM) is estimated from primary organic carbon (OC) using spatially and seasonally varying OM/OC ratios at 0.1° x 0.1° resolution (Philip et al., 2014b). The thermodynamic equilibrium model ISORROPIA-II (Fountoukis and Nenes, 2007), implemented by Pye et al. (2009), is used to calculate gas-aerosol partitioning. Total PM$_{10}$ is calculated following van Donkelaar et al. (2010), but at 40% RH here for consistency with the
IMPROVE network gravimetric analysis in the range of 30-50% RH (Solomon et al., 2014). Particle scatter and aerosol optical depth are calculated at modelled ambient RH based on dry species mass concentrations and aerosol physical and optical properties. The GEOS-Chem aerosol simulation has been extensively evaluated with observations of mass (van Donkelaar et al., 2015; Li et al., 2016), composition (Achakulwisut et al., 2017; Kim et al., 2015; Marais et al., 2016; Philip et al., 2014a; Ridley et al., 2017; Zhang et al., 2013), and scatter (Drury et al., 2010).

We conduct a simulation for the year 2006, to represent the period of greatest measurement density of collocated b$_{sp}$ and PM sites over North America. We archive model fields every hour over North America. We simulate PM$_{10,}$ PM$_{2.5}$ and b$_{sp}$, allowing for the comparison of model mass scattering efficiency coincident with that measured at IMPROVE network sites over the same time period over North America.

### 2.3 Determining mass scattering efficiciency ($\alpha_{sp}$)

One method of determining mass scattering efficiencies from measurements involves $b_{sp}$ measurements and particle mass concentration measurements ($M_{meas}$). Mass scattering efficiency of a given aerosol population can be defined as the ratio of particle scatter to mass.

$$\alpha_{sp,meas} = \frac{b_{sp,meas}}{M_{meas}} \tag{1}$$

Hourly mass scattering efficiencies are determined using collocated measurements of $b_{sp}$ and mass concentrations from the IMPROVE network, treating IMPROVE mass concentrations as constant over each 24h sampling period. Total scatter is typically dominated by fine mode aerosols, but in certain conditions coarse dust can also make a significant contribution (White et al., 1994). Thus, measured $PM_{10}$ mass is used in the denominator of Eq. (1).

Multiple definitions of $\alpha_{sp}$ exist. We define $\alpha_{sp}$ operationally here based on optical measurements at ambient RH, and PM measurements at controlled RH (treated as 40% RH for consistency with IMPROVE protocols prior to 2011). At 40% RH, hygroscopic components of $PM_{10}$ will have associated water, and thus measured $PM_{10}$ mass is not treated as dry. We compare these measured $\alpha_{sp}$ with calculated $\alpha_{sp}$ based on species specific mass scattering efficiencies ($\alpha_{GC,j}$) used in GEOS-Chem, constrained with mass concentrations ($M_j$) and $PM_{10}$ mass measured by IMPROVE.

$$\alpha_{sp,calc} = \frac{b_{sp,calc}}{PM_{10,meas}} = \frac{\sum_j \alpha_{GC,j} M_j}{PM_{10,meas}} \tag{2}$$

To reduce the impacts of meteorological variation on the comparison of measured and calculated mass scattering efficiency, we perform averages of hourly $b_{sp,calc}$, $b_{sp,meas}$, and $PM_{10}$ over the entire sampling period at each IMPROVE site $i$. Eq. (3) is then used to obtain average calculated and measured mass scattering efficiency at each site.

$$\alpha_{sp,avg,i} = \frac{b_{sp,avg,i}}{PM_{10,avg,i}} \tag{3}$$

Although the OPTEC open air nephelometer reduces truncation error compared with other nephelometers, truncation error can be significant for coarse particles (Hand and Malm, 2007; Lowenthal and Kumar, 2006). Thus our analysis below focuses on conditions dominated by fine mode aerosols, and mechanisms affecting fine mode aerosols.

Appendix A describes the calculation of mass scattering efficiency in more detail. This approach enables isolation of the mass scattering efficiencies used in GEOS-Chem from the species concentrations.

### 2.4 Introducing an alternate hygroscopic growth scheme

We examine for GEOS-Chem the use of a widely adopted alternate hygroscopic growth scheme, in which aerosol hygroscopic growth is defined by a single parameter, $\kappa$ (Petters and Kreidenweis 2007, 2008, 2013). This representation of water uptake by aerosols was originally developed for supersaturated CCN conditions, but in recent years has been used extensively in subsaturated conditions (Dusek et al., 2011; Hersey et al., 2013).

The hygroscopic parameter $\kappa$ is defined by

$$\frac{1}{a_w} = 1 + \kappa \frac{V_d}{V_w} \qquad (4)$$

where $V_d$ is dry particulate matter volume, $V_w$ is the water volume and $a_w$ is water activity (Petters and Kreidenweis, 2013), which is unity for secondary inorganic aerosols (SIA) and organic aerosols (OA). The diameter growth factor (GF=D/D$_d$) can be expressed (Snider et al., 2016) as

$$GF = \left(1 + \kappa \frac{RH}{100-RH}\right)^{1/3} \qquad (5)$$

where D is the wet aerosol radius and $D_d$ is the dry aerosol radius. Typically, $\kappa$ is in the range of 0.5-0.7 for SIA (Hersey et al., 2013; Kreidenweis et al., 2008; Petters & Kreidenweis, 2007) and 0-0.2 for OA (Duplissy et al., 2011; Kreidenweis et al., 2008; Rickards et al., 2013; Snider et al., 2016).

## 3. Results

### 3.1 Understanding the current representation of $\alpha_{sp}$

Figure 2 (left) shows measured vs. calculated mass scattering efficiency using GEOS-Chem default optical tables. Each point represents the average $\alpha_{sp}$ over the entire sampling period at each IMPROVE site. A significant correlation (r=0.94) is apparent, however a bias in $\alpha_{sp}$ is evident. A positive correlation between average mass scattering efficiency and RH is apparent; sites with low average RH have low average $\alpha_{sp}$ and vice versa. (The right panel of Figure 2 is discussed below.)

To further investigate the RH dependence of this bias, we separate our analysis of calculated $\alpha_{sp}$ into 3 relative humidity groupings: 0-35% (low), 35-65% (mid) and 65-95% (high). The IMPROVE data is divided among the RH groupings using IMPROVE measurements of hourly RH. Within each grouping, average calculated and measured mass scattering efficiencies are obtained for each site using Eq. (3). The blue dots in Fig. 3 show average calculated vs measured $\alpha_{sp}$ for each RH range. In the low RH case, a significant overestimation of mass scattering efficiency is apparent at most sites, with a bias of 82% indicated by the slope. In the mid RH case, overestimation of $\alpha_{sp}$ is less significant but still apparent with a bias of 40% indicated by the slope. At high RH, bias is weak.

To further understand the source of the bias in calculated mass scattering efficiency, we now examine calculated $\alpha_{sp}$ in conditions dominated by different aerosol types. Using IMPROVE measurements of 24 hr PM$_{2.5}$ mass and speciation and PM$_{10}$ mass, the IMPROVE data is grouped based on dominant aerosol type. Within each group, average calculated and measured mass scattering efficiency is obtained for each site using Eq. (3). Figure 4 shows in blue average measured vs calculated $\alpha_{sp}$ using default optical tables for conditions where measured PM$_{2.5}$ is dominated (>60%) by secondary inorganic aerosol, organic aerosol and fine dust, as well conditions where PM$_{10}$ is dominated (>60%) by PM$_{coarse}$ (PM$_{10}$-PM$_{2.5}$). The scatterplot in the SIA dominant case resembles the overall relationship shown in Fig. 2. $\alpha_{sp}$ is overestimated at most sites, with significant correlation (r=0.88), and a bias evident in the offset of 0.70. Where OA is the dominant component of PM$_{2.5}$,

the slope is close to unity (1.02) but the large offset of 0.80 $m^2 g^{-1}$ results in $\alpha_{sp}$ being largely overestimated. Where dust is the dominant fine aerosol, correlation is significant (r=0.89) and mass scattering efficiency is accurately calculated at the vast majority of sites, despite a prominent outlier at a site in the Columbia River Gorge, Washington. The $PM_{coarse}$ dominant case shows significant correlation (r=0.88) and a slight tendency for overestimation of $\alpha_{sp}$. As this case is not independent from the other cases, this overestimation is likely linked to the overestimation in the OA and SIA dominant cases as demonstrated below.

These results indicate that the bias in calculated mass scattering efficiency arises mostly due to the representation of the physical and optical properties of secondary inorganic and organic aerosols. The following will focus on improving the representation of physical and optical properties of these two aerosol types.

**3.2 Changing the physical properties of SIA and OA**

Figure 5 shows mass scattering efficiency as a function of aerosol size for secondary inorganic (orange) and organic (blue) aerosols for dry aerosols (solid) and aerosols at 80% RH (dashed lines) as calculated using a Mie algorithm ((Mishchenko et al., 1999). Water uptake at 80% RH for OA and SIA is calculated using default hygroscopic growth factors from GEOS-Chem. The uptake of water increases aerosol scatter, decreases aerosol density and decreases the refractive index. The increase in aerosol scatter with increasing ambient RH drives the increase in $\alpha_{sp}$.

The points in Fig. 5 represent the current mass scattering efficiency values of OA and SIA in GEOS-Chem. For dry aerosols, $\alpha_{sp}$=4.4 $m^2 g^{-1}$ for OA and $\alpha_{sp}$=3.2 $m^2 g^{-1}$ for SIA. In a review of ground-based estimates of aerosol mass scattering efficiencies, Hand et al. (2007) found dry $\alpha_{sp}$ values of 2.5 $m^2 g^{-1}$ for ammonium sulfate, 2.7 $m^2 g^{-1}$ for ammonium nitrate, and 3.9 $m^2 g^{-1}$ for particulate organic matter. These values suggest that the default optical tables in GEOS-Chem currently overestimate mass scattering efficiency of SIA and OA in dry conditions. This reaffirms the overestimation of $\alpha_{sp}$ in dry conditions evident in the left panel of Fig. 3. As aerosol size is the strongest determinant of dry mass scattering efficiency, we begin by examining the dry sizes of SIA and OA in GEOS-Chem.

The current dry sizes of SIA and OA in GEOS-Chem were informed by measurements from several aircraft campaigns over eastern North America during the summer of 2004 (Drury et al., 2010) as part of the The International Consortium for Atmospheric Research on Transport and Transformation (ICARTT) (Fehsenfeld et al., 2006; Singh et al., 2006). Aerosol surface area and volume distributions fluctuate seasonally in the North Eastern U.S., with summer maxima and winter minima (Stanier et al., 2004). We divide our analysis at low RH by season, in an effort to discern a seasonal pattern in the overestimation of $\alpha_{sp}$.

Figure 6 (blue) shows seasonal measured vs. calculated mass scattering efficiency in dry conditions using default optical tables (Table A1). Estimations of $\alpha_{sp}$ are most accurate in the summer, consistent with the dry sizes chosen by Drury et al. (2010) which were informed by summertime size distribution measurements. The larger overestimation of $\alpha_{sp}$ in all

other seasons, most notably in winter, is consistent with the seasonality in aerosol size distributions observed by Stanier et al. (2004).

### 3.2.1 Efflorescence relative humidity

To address the overestimation of mass scattering efficiency in dry conditions illustrated in Fig. 3 and Fig. 6, we begin by accounting for efflorescence transitions in secondary inorganic aerosols. Efflorescence phase transitions are characterized by nucleation of the crystalline phase followed by rapid evaporation of water. Field measurements have found evidence for these transitions (Martin et al., 2008). The efflorescence relative humidity (ERH) of ammonium sulfate reported in several experimental studies range from 35-40% (Ciobanu et al., 2010). Laboratory tests have shown that mixtures of sulfate-nitrate-ammonium particles will undergo efflorescence when the ammonium sulfate fraction is high (Dougle et al., 1998; Martin et al., 2003). This condition is true at most global measurement sites, with the possible exception of Europe, where particles are nitrate rich (Martin et al., 2003).

We therefore define the hygroscopic growth factor for SIA as unity for RH ≤ 35%, linearly increasing between 35-40% RH from unity to $GF_{40\%}$ (calculated by Eq. (5)), and following the default (or κ-Kohler) growth curve for RH ≥ 40%. Incorporating an ERH for SIA and consequently inhibiting hygroscopic growth of SIA below 35% RH significantly reduces the overestimation of mass scattering efficiency in dry conditions. In the case of default hygroscopic growth in GEOS-Chem, the overall dry bias in $\alpha_{sp}$ is reduced from 82% to 48%.

### 3.2.2 Aerosol dry size

To address the remaining overestimation of mass scattering efficiency in dry conditions we explore different dry sizes of secondary inorganic and organic aerosols. Effective variance may also be important (Chin et al., 2002) but given insufficient information to simultaneously constrain size and variance, we focus on size. Figure 7 shows the slope of the average measured vs calculated $\alpha_{sp}$ plot for RH<35% for dry radii ranging from 0.048 to 0.074 μm at intervals of 0.001 μm, assuming SIA and OA have the same dry size. The slope of the best fit line acts as an indicator of the appropriate dry size for each season. Sensitivity tests exploring alternative error metrics (RMSE, MSE) yielded similar results. The slope decreases steadily as dry radius is decreased in all seasons. Using the dry radius which gives a slope of unity, we find that aerosols are largest in summer (r=0.067 μm), smallest in winter (r=0.051 μm), and in between in spring and fall (0.059 μm and 0.054 μm, respectively). The spring and summer radii are consistent with accumulation mode size distribution measurements performed by Levin et al. (2009) in the spring and summer of 2006. Averaging the sizes from all four seasons results in an annual representative dry radius of 0.058 μm. This annual radius is smaller than the GEOS-Chem default sizes of SIA and OA that were informed by summertime measurements alone (Drury et al., 2010).

Figure 6 (red) shows seasonal measured vs. calculated $\alpha_{sp}$ in dry conditions using new representative annual geometric mean radius of 0.058 μm for SIA and OA. This change in geometric mean radius reduces the overestimation of

$\alpha_{sp}$ in all seasons, with the largest improvement in fall (slope decreases from 1.84 to 1.17) and winter (slope decreases from 1.94 to 1.20). Changes in correlation are minor. For the remainder of the analysis, this new dry radius of 0.058 μm is implemented for SIA and OA.

### 3.2.3 Aerosol hygroscopicity

We now examine the implementation of the widely adopted κ-Kohler hygroscopic growth scheme described in section 2.4. A range of measured κ values for SIA ($\kappa_s$) and OA ($\kappa_o$) exist in the literature. We explore the range of possible κ values, using the slope of the measured vs calculated $\alpha_{sp}$ plot as an indicator of the appropriate values.

      Figure 8 shows the slope of the measured vs calculated $\alpha_{sp}$ plot for κ values for SIA ($\kappa_s$) ranging from 0.5-0.7 and for OA ($\kappa_o$) ranging from 0.08-0.20. Slope increases steadily as $\kappa_s$ and $\kappa_o$ increase. A slope of unity identifies representative

values of $\kappa_s$=0.61 and $\kappa_o$=0.10. These values are in the middle of the range of measured κ values (Duplissy et al., 2011; Hersey et al., 2013; Kreidenweis et al., 2008; Petters and Kreidenweis, 2007; Rickards et al., 2013).

      Figure 9 shows the diameter growth factor as a function of relative humidity following κ-Kohler theory, as well as GADS hygroscopic growth for both SIA and OA used in the default GEOS-Chem model. Hygroscopic growth from the Aerosol Inorganic Model (AIM) at T=298 K (Wexler and Clegg, 2002) and laboratory measurements (Wise et al., 2003) are

also shown for ammonium sulfate (Snider et al., 2016). The GADS hygroscopic growth schemes used in the default GEOS-Chem simulation are characterized by larger growth at low RH and smaller growth at high RH for both secondary inorganic and organic aerosols. The κ-Kohler scheme exhibits greater consistency with both AIM and laboratory hygroscopic growth for SIA.

      Using the revised dry size of 0.058 μm and the κ-Kohler theory of hygroscopic growth, we calculate revised

physical and optical properties for SIA and OA over a range of RH values. Table A1 contains geometric mean radius, extinction efficiency and single scattering albedo for the revised optical tables at 8 relative humidity values.

      Figure 2 (right) shows the measured vs calculated mass scattering efficiency using these revised optical tables for SIA and OA. The overestimation of mass scattering efficiency has been eliminated with these revised aerosol properties, with a slope of 1.00 and an offset of 0.09. Correlation remains significant at r=0.96.

25       Figure 4 (red) shows measured vs calculated $\alpha_{sp}$ in conditions dominated by different aerosol types using the revised optical tables. The overestimation of $\alpha_{sp}$ in SIA dominant conditions using the default optical tables has been eliminated, with a slope of 1.03 and a decreased offset (0.70 to 0.1). The large overestimation of $\alpha_{sp}$ that was apparent in OA dominant conditions has been reduced by a factor of 2. $\alpha_{sp}$ remains accurately estimated at the majority of dust dominant sites, with the outlier at the Columbia River Gorge site in Washington still skewing the best fit line. The slight

overestimation of $\alpha_{sp}$ that was present in the $PM_{coarse}$ dominant case using default optical tables has been eliminated using the revised tables (offset 0.33 to 0.03). Slight increases in correlation coefficients are apparent in all cases except for the SIA dominant case, where it decreased by 0.02.

Figure 3 (red) shows measured vs calculated $\alpha_{sp}$ using revised optical tables. The overestimation in $\alpha_{sp}$ has been significantly reduced in the low RH case (slope=1.82 to slope=1.09) and in the mid RH case (slope=1.40 to slope=1.01) compared to when default optical tables were used. The slight overestimation in high RH conditions present in the default case has also been reduced, as shown by the decreased offset (0.90 to 0.71).

**3.3 Changes in GEOS-Chem simulated $\alpha_{sp}$**

Here, we examine how these changes to aerosol properties impact both GEOS-Chem simulation of mass scattering efficiency over North America and the fit between modeled and measured $\alpha_{sp}$ at IMPROVE sites. These simulations rely on GEOS-Chem simulations of aerosol composition using GEOS RH fields.

Figure 10 shows the relative and absolute change in mass scattering efficiency when switching from the default to revised optical tables. Continental mean $\alpha_{sp}$ increased by 16%. Increases in $\alpha_{sp}$ range from 25-45% in northeastern regions of North America, corresponding to an increase of 1.5-3.5 $m^2 g^{-1}$. These larger changes reflect the higher RH and SIA fractions. Decreases in $\alpha_{sp}$ of up to 15% or -0.5 $m^2 g^{-1}$ are found in the southwest where RH is low and mineral dust dominates.

Figure 11 shows GEOS-Chem annual average mass scattering efficiency using default (top) and revised (bottom) optical tables over North America for the year 2006. The overlaying circles represent average measured $\alpha_{sp}$ at IMPROVE network sites for the year 2006, and the outer rings show the coincident simulated $\alpha_{sp}$ for each site. We exclude sites within 1° of the coast where sea-salt affects $\alpha_{sp}$, as well as sites where elevation differs from average gridbox elevation by more than 1500 meters. These criteria result in a decrease from 24 to 19 in the number of sites available for the analysis in 2006.

Using default optical tables, simulated continental mean $\alpha_{sp}$ is 5.4 $m^2 g^{-1}$. A maximum $\alpha_{sp}$ of 10 $m^2 g^{-1}$ occurs in British Columbia, and a minimum $\alpha_{sp}$ of 1.7 $m^2 g^{-1}$ occurs in the southwest United States. Using revised optical tables, simulated continental mean $\alpha_{sp}$ is 6.3 $m^2 g^{-1}$, with a maximum of 12.5 $m^2 g^{-1}$ in the northwest, and a minimum of 1.5 $m^2 g^{-1}$ in the southwest. The elevated mass scattering efficiencies in the northwest can be attributed in part to the high average RH in this region of 83%.

Figure 12 (left) shows coincident measured vs simulated mass scattering efficiency at the 19 IMPROVE sites, using default optical tables. Correlation is significant (r=0.88), but a bias in simulated $\alpha_{sp}$ is apparent (slope=0.83). Simulated $\alpha_{sp}$ is notably biased low at sites in the southeastern United States where average $\alpha_{sp}$ is largest, and simulated $\alpha_{sp}$ is notably biased high at sites in the southwestern United States where average mass scattering efficiency is lowest. Sites with lowest average RH correspond to those with the lowest average mass scattering efficiency and vice versa. The tendency of mass scattering efficiency to be overestimated at low RH reflects the tendency that was originally seen in Fig. 4.

Figure 12 (right) shows coincident measured vs simulated $\alpha_{sp}$ using revised optical tables. Correlation remains significant (r=0.89), and a decrease in bias is evident from the increase in slope (0.83 to 0.93) and decrease in offset (0.47 to

0.08). Most sites now lie closer to the 1:1 line. The overestimation of simulated $\alpha_{sp}$ in the southwest, where RH is low, has been reduced or eliminated at all sites.

### 3.4 Comparison with AERONET measurements

Appendix B investigates changes to simulated AOD, and compares measured and simulated AOD at AERONET sites. Although large relative increases upwards of 60% in average AOD are evident in large parts of northern high-latitudes where absolute AOD is small, absolute AOD generally changes by less than 0.1 (Fig. B1). Comparisons with AERONET AOD reveal that the revised optical properties slightly improve the simulation of AOD worldwide (slope decreases from 1.08 to 1.00) despite the large influence of other factors (e.g. ambient aerosol concentrations) upon AOD.

### 4. Conclusions

The current representation of mass scattering efficiency in the GEOS-Chem global chemical transport model was evaluated using collocated ground-based measurements of particle mass, speciation, scatter and relative humidity from the IMPROVE network.

Calculated mass scattering efficiency had a positive bias using default physical and optical properties used in the GEOS-Chem model. This bias was most significant when $PM_{2.5}$ mass was dominated by secondary inorganic (SIA) or organic aerosols (OA). Mass scattering efficiency in $PM_{2.5}$ dust and coarse particulate matter dominant conditions was accurately represented at the majority of IMPROVE sites.

Relative humidity played an important role in the severity of the bias in mass scattering efficiency. Mean $\alpha_{sp}$ was overestimated by 82% in dry conditions (RH<35%). This bias was largest in the winter (94%) and smallest in the summer (27%). Implementing an efflorescence relative humidity for SIA and thus inhibiting hygroscopic growth below 35% RH decreased the dry bias by 34%. An annual representative dry geometric mean radius of 0.058 μm for SIA and OA decreased the dry mass scattering efficiency of these aerosols, and subsequently further reduced the bias in dry conditions to 9%.

κ-Kohler theory was implemented for the hygroscopic growth of SIA and OA, which is characterized by smaller growth factors at low RH and larger growth factors at high RH compared to default growth factors in GEOS-Chem. κ values of 0.61 for SIA and 0.10 for OA eliminated the overall bias in calculated mass scattering efficiency.

These changes to SIA and OA optical tables resulted in a continental mean increase in GEOS-Chem simulated mass scattering efficiency of 16%. Northeastern regions of North America exhibited the largest increases (25-45%) due to high RH and SIA fractions, while southwestern regions of the continent exhibited decreases in $\alpha_{sp}$ of up to 15% due to low RH and high dust fractions. These changes to the GEOS-Chem optical tables improved the fit between measured and simulated mass scattering efficiency at IMPROVE sites, reflected in the changes to the slope (0.83 to 0.93) and the offset (0.47 to 0.08).

Future work should examine the implications of these changes for satellite-derived estimates of fine particulate matter that depend on the relationship of AOD with $PM_{2.5}$. Future work should also expand analysis of the representation of mass scattering efficiency for other years, and by incorporating measurements from other ground based measurement networks such as the Surface PARTiculate MAtter network (SPARTAN), which provides measurements of particulate mass,

speciation and scatter in populated regions worldwide (Snider et al., 2015; Snider et al., 2016). Such comparisons may also be useful to evaluate and improve prognostic simulations of aerosol size (Mann et al., 2010; Spracklen et al., 2005; Trivitayanurak et al., 2008; Yu and Luo, 2009) . Representation of particle RH history may also be important (Wang et al., 2008).

**Appendix A**

**A.1 $b_{sp}$ and $\alpha_{sp}$ calculations in GEOS-Chem**

In GEOS-Chem, surface level $b_{sp}$ is calculated using model particle mass concentrations and local relative humidity, as well as predefined mass densities and aerosol optical properties for each aerosol component following:

$$b_{sp} = \sum_{species,i} \frac{\frac{3}{4}*\left(\frac{R_{w,i}}{R_{d,i}}\right)^2 *M_{d,i}*Q_{w,i}*SSA_{w,i}}{\rho_{d,i}*R_{d,i}} \tag{A1}$$

where $\rho_d$ is the dry particle mass density, $R_w$ is the effective radius (defined as the ratio of the third to second moment of an

aerosol size distribution), $R_d$ is the dry effective radius, $M_d$ is the dry surface level mass concentration, $Q_w$ is the extinction efficiency, and $SSA_w$ is the single scattering albedo. Parameters with the subscript w indicate values at ambient RH. Species included in this calculation are $SO_4^{2-}$, $NH_4^+$, $NO_3^-$, BC, OM and fine and coarse dust and sea salt.

Dividing Eq. (A1) by total surface level $PM_{10}$ results in the following equation for mass scattering efficiency

$$\alpha_{sp} = \frac{B_{sp}}{PM_{10}} = \frac{\sum_{species,i} \frac{\frac{3}{4}*\left(\frac{R_{w,i}}{R_{d,i}}\right)^2 *\frac{M_{d,i}}{PM_{10}}*Q_{w,i}*SSA_{w,i}}{\rho_{d,i}*R_{d,i}}}{PM_{10}} \tag{A2}$$

The effective radius, extinction efficiency and single scattering albedo in Eq. (A1) and (A2) are obtained from GEOS-Chem optical tables for the ambient RH values measured by IMPROVE. Dry mass density $\rho_d$ is specified for each aerosol species in GEOS-Chem (Table A2). $M_{d,i}$ and $PM_{10}$ are obtained from IMPROVE network measurements of aerosol mass and composition. $\alpha_{sp}$ calculated by Eq. (A2) is compared to $\alpha_{sp}$ directly measured by the IMPROVE network.

Mass scattering efficiency is dependent on particle density, refractive index and particle size. Mass scattering

efficiency is typically most dependent on aerosol size, which is dictated by both the dry size distribution chosen to represent a given aerosol species, and by the hygroscopic growth scheme used to represent aerosol water uptake for hydrophilic species.

**A.2 Incorporating IMPROVE Network Measurements**

The IMPROVE network measures every three days $PM_{2.5}$ mass and speciation and $PM_{10}$ mass. The IMPROVE particle sampler consists of four independent modules with separate inlets and pumps. The first three modules (A, B and C) collect only fine particulate matter ($PM_{2.5}$), while the $4^{th}$ module (D) collects both fine and coarse particles ($PM_{10}$). Module A
collects $PM_{2.5}$ on a Teflon filter, which undergoes gravimetric analysis for total $PM_{2.5}$ mass and x-ray florescence for elemental concentrations (including Al, Si, Ca, Fe, Ti). The nylon filter in module B undergoes ion chromatography analysis for $SO_4^{2-}$, $NO_3^-$, $NO_2^-$ and $Cl^-$. Module C contains a quartz filter that is analyzed for organic and elemental carbon via thermal optical reflectance. The Teflon filter in module D undergoes gravimetric analysis for $PM_{10}$ mass (Malm et al., 1994; Malm et al., 2004). Prior to gravimetric analysis, filters A and D undergo equilibration at 30-50% RH and 20-25 °C for several
minutes (Solomon et al., 2014).

   The GEOS-Chem model partitions OM into hydrophilic and hydrophobic fractions, so the same is done for OM measured by IMPROVE to enable isolation of mass scattering efficiency in our comparisons. OM in remote regions tends to be highly oxidized, and oxidation level of organics has been shown to positively correlate with hygroscopicity (Duplissy et al., 2011; Jimenez et al., 2009; Ng et al., 2010). We treat measured OM as 90% hydrophilic, due to the rural nature of
IMPROVE sites. EC is treated as 50% hydrophilic. As speciation of coarse material is unavailable, we treat all coarse material as crustal in origin, an assumption that may breakdown at coastal sites. We partition fine and coarse dust measured by the IMPROVE network into the GEOS-Chem size bins using the dust particle size distribution (PSD) described by Zhang et al. (2013).

**Appendix B**

The *Ae*rosol *Ro*botics *Net*work (AERONET) is a long-term network of ground based sun photometers that provides continuous, cloud-screened measurements of aerosol optical depth (AOD) at several fixed wavelengths in the visible and near infrared (Holben et al., 1998). The calculation of AOD in GEOS-Chem is performed using simulated mass concentrations of aerosol species and mass extinction efficiencies, summed over all vertical layers. Our analysis of mass scattering efficiency can therefore be extended globally by comparing GEOS-Chem calculated AOD to AOD measured at
AERONET sites. During our simulation year of 2006, AERONET consisted of 231 sites across the globe.

   Here we examine how the changes to SIA and OA properties impact GEOS-Chem simulated AOD globally. Figure B1 shows the relative (top) and absolute (bottom) changes in AOD. Global mean AOD increases by 19%. Relative changes in AOD are most pronounced in Northern regions where mean relative humidity is high, with increases in simulated AOD ranging from 50-90%. Decreases in AOD between 0-20% are present in most of the southern hemisphere, in part due to the
lower average RH. Absolute changes in AOD show a similar tendency, with slight increases in AOD of up to 0.2 in northern regions, and slight decreases of up of -0.09 in southern regions. An exception to this is seen over parts of China, where AOD increases by 0.5 due to the elevated SIA and OA concentrations.

Figure B2 shows coincident measured (inner circles) and simulated (outer rings) AOD for the year 2006 using default optical tables (top) and revised optical tables (bottom). We exclude sites within 1° of the coast, as well as sites where elevation differs from average gridbox elevation by more than 1500 meters. We also exclude sites where average $PM_{2.5}$ is dominated by dust (dust/$PM_{2.5}$>0.6), to focus on the representation of the optical properties of SIA and OA. Across the globe, we see that AOD is both over and underestimated. AOD is overestimated at most sites in Africa, with the most notable overestimation at the site in Nigeria. AOD is moderately overestimated at sites in Australia. Underestimation of AOD occurs at most sites in South America, as well as at sites in southern North America and southern Asia.

Figure B3 shows coincident measured vs simulated AOD at AERONET sites for default (left) and revised (right) optical tables. The correlation coefficient (r=0.80 to r=0.78) changes insignificantly, while the slope decreases from 1.08 to 1.00 when switching to the revised optical tables. In summary, the revised optical properties developed for North America slightly improve the representation of AOD at the global scale, despite the large influence of other factors (e.g. ambient aerosol concentrations and composition) upon AOD.

**Data Availability**

IMPROVE network data for 2000-2010 can be accessed at http://vista.cira.colostate.edu/Improve/improve-data/.

**Acknowledgements**

Research described in this article was conducted under contract to the Health Effects Institute (HEI), an organization jointly funded by the United States Environmental Protection Agency (EPA) (Assistance Award No. R-82811201) and certain motor vehicle and engine manufacturers. The contents of this article do not necessarily reflect the views of HEI, or its sponsors, nor do they necessarily reflect the views and policies of the EPA or motor vehicle and engine manufacturers.

IMPROVE is a collaborative association of state, tribal, and federal agencies, and international partners. US Environmental Protection Agency is the primary funding source, with contracting and research support from the National Park Service. The Air Quality Group at the University of California, Davis is the central analytical laboratory, with ion analysis provided by Research Triangle Institute, and carbon analysis provided by Desert Research Institute.

We thank Environment and Climate Change Canada for providing nephelometer data at the site in Egbert, Ontario.

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

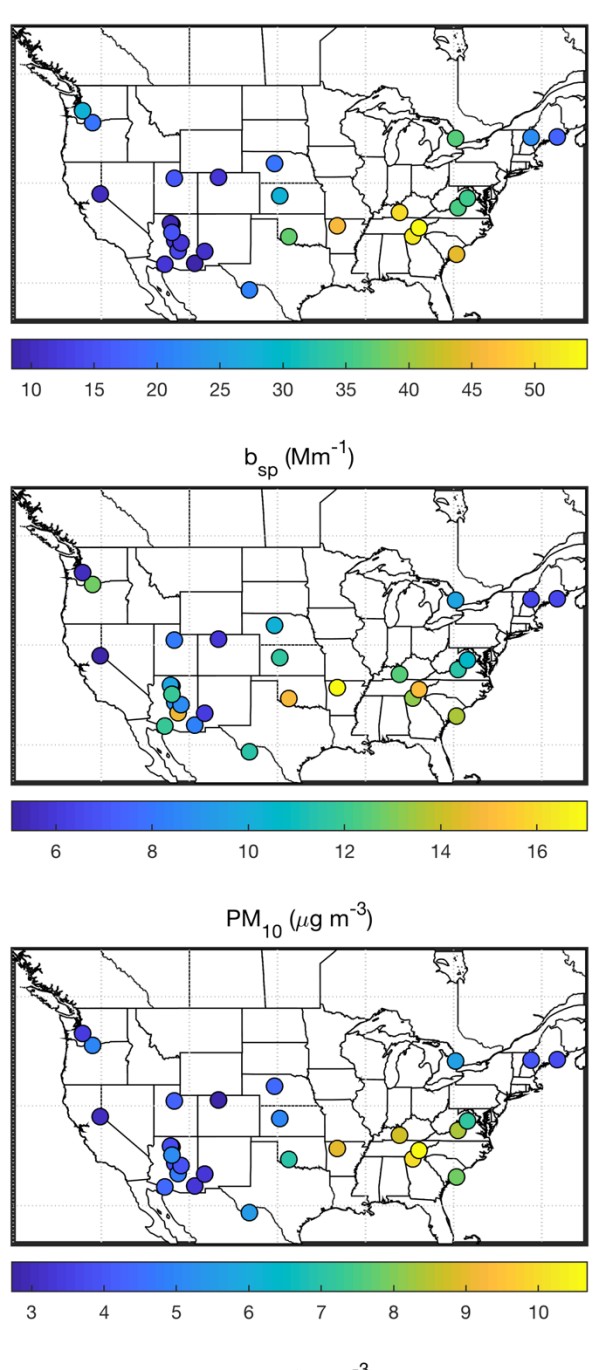

**Figure 1: Map of IMPROVE sites with collocated scatter ($b_{sp}$) at 550 nm and ambient relative humidity, $PM_{10}$ and $PM_{2.5}$ measurements in North America between 2000-2010.**

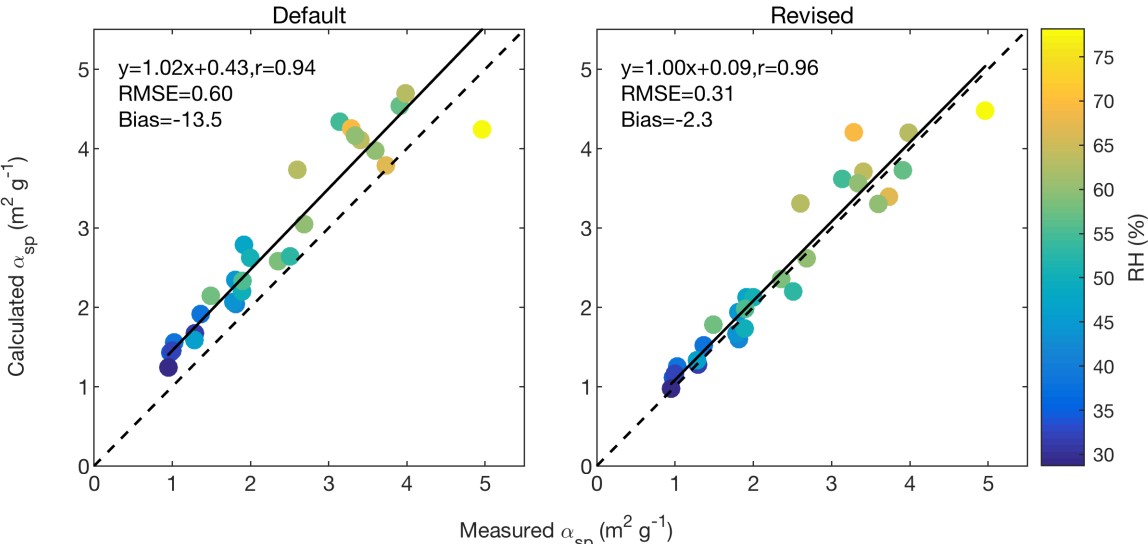

**Figure 2: Average measured vs calculated $\alpha_{sp}$ at 550 nm at IMPROVE sites between 2000-2010 using GEOS-Chem default optical tables and revised optical tables. The colour of each point corresponds to the average relative humidity at the site. The 1:1 line is black. Slope, offset and correlation coefficient are inset.**

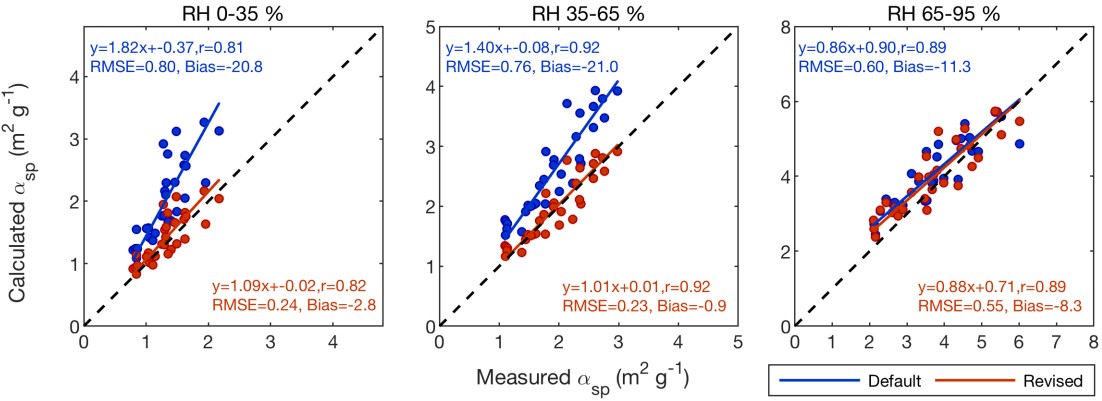

**Figure 3: Average measured versus calculated $\alpha_{sp}$ at 550 nm at IMPROVE sites between 2000-2010 using GEOS-Chem default and revised optical tables (Table A1) for measurements taken in 0-35 % RH, 35-65 % RH and 65-95 % RH conditions. The 1:1 line is black. Slope, offset and correlation coefficient are inset.**

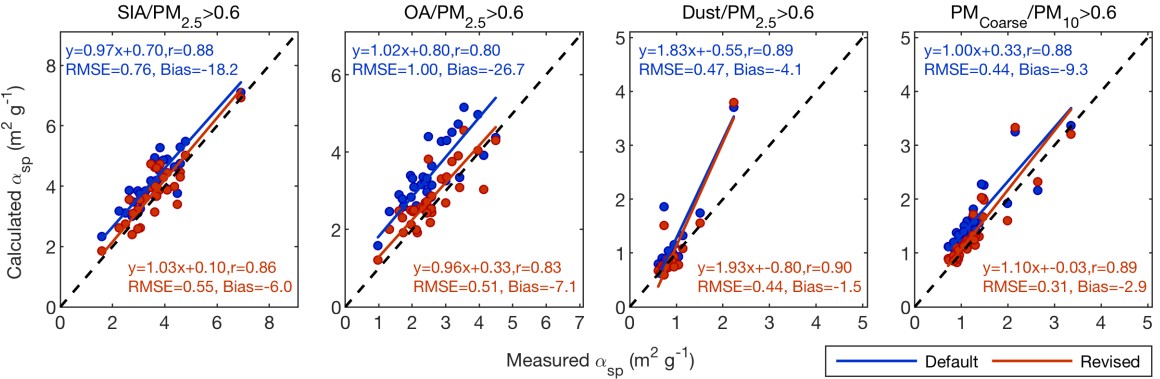

**Figure 4: Average measured versus calculated $\alpha_{sp}$ (550 nm) at IMPROVE sites between 2000-2010 using GEOS-Chem default and revised optical tables for measurements taken in conditions dominated by secondary inorganic aerosols (SIA), organic aerosols (OA), fine dust, and PM$_{coarse}$ (PM$_{10}$-PM$_{2.5}$). The 1:1 line is black. Slope, offset, and correlation coefficient are inset.**

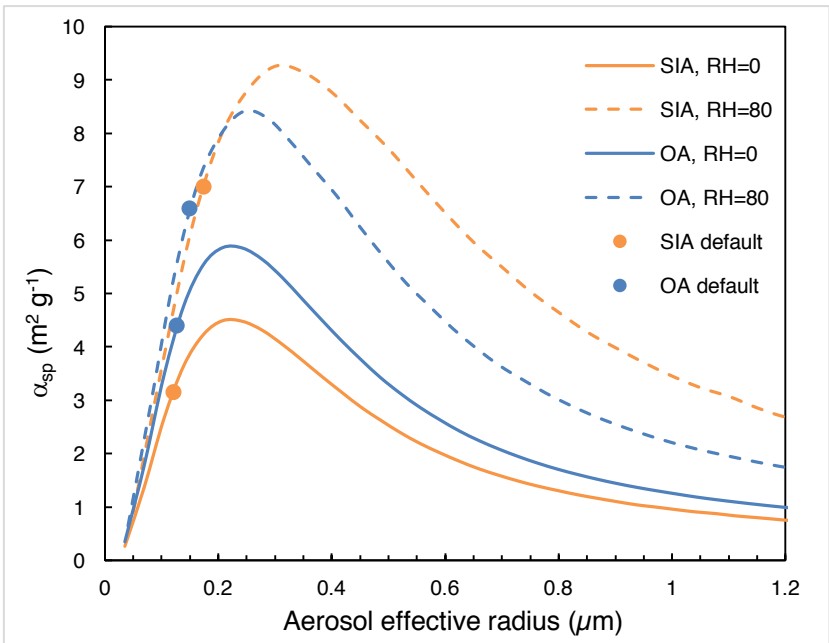

**Figure 5: Mass scattering efficiency ($\alpha_{sp}$) at 550 nm as a function of aerosol wet effective radius for organic aerosol and secondary inorganic aerosol. Solid lines show $\alpha_{sp}$ for dry aerosol (RH=0%), dashed lines show $\alpha_{sp}$ for aqueous aerosols (RH=80%). Points represent the default size in GEOS-Chem.**

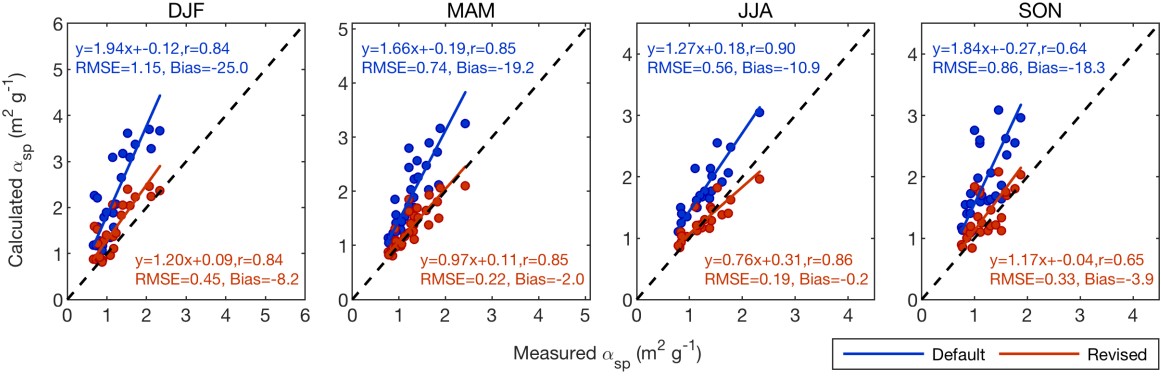

**Figure 6: Average measured versus calculated $\alpha_{sp}$ (550 nm) at IMPROVE sites between 2000-2010 using GEOS-Chem default and revised optical tables for measurements taken in dry conditions (RH<35 %) in winter, spring, summer and fall. The 1:1 line is black. Slope, offset and correlation coefficient are inset.**

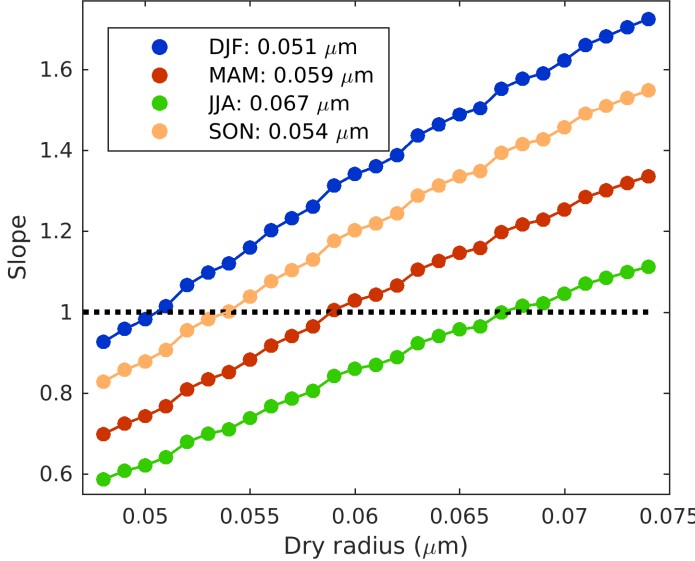

**Figure 7: Slope of measured vs. calculated $\alpha_{sp}$ plot versus dry geometric mean aerosol radius, by season. Winter (DJF) is in blue, spring (MAM) in red, summer (JJA) in green and fall (SON) in orange. The line Slope=1 is shown in black. Numbers in the legend represent the dry radius for which slope=1 for each season.**

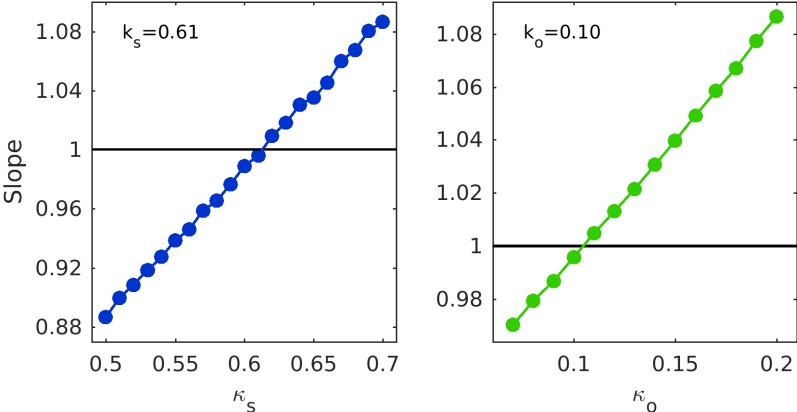

**Figure 8: Slope of measured vs. calculated $\alpha_{sp}$ plot as a function of the $\kappa$ of secondary inorganic aerosols ($\kappa_S$, left) and the $\kappa$ of organic aerosols ($\kappa_O$, right). The line slope=1 is shown in black. $\kappa_s$ and $\kappa_o$ values for which slope=1 are inset.**

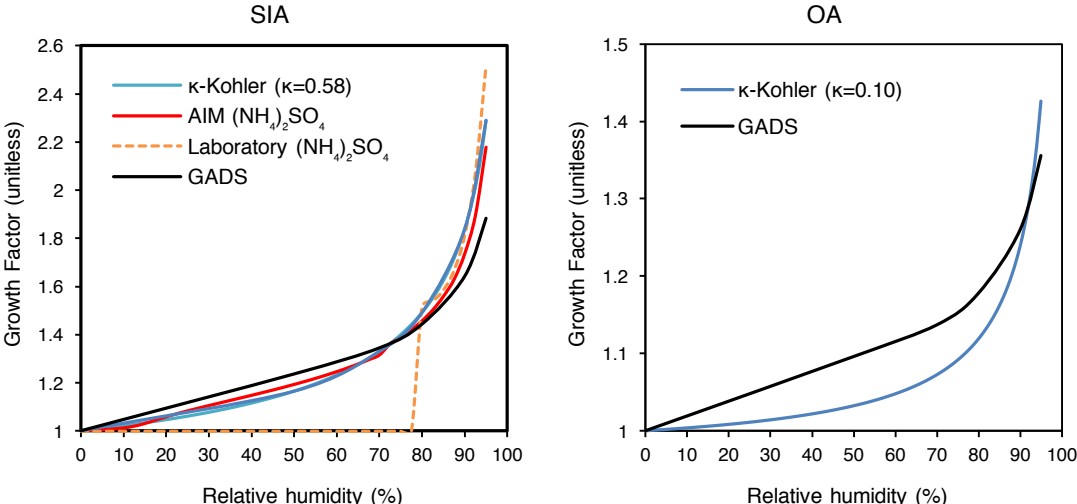

**Figure 9: Hygroscopic growth factor curves for secondary inorganic aerosols (SIA, left) and organic aerosols (OA, right). GADS (Global Aerosol Data Set) hygroscopic growth from empirical data and $\kappa$-Kohler hygroscopic growth are shown for both SIA and OA. For ammonium sulfate, AIM (Aerosol Inorganic Model) hygroscopic growth at T=298 K (Wexler and Clegg, 2002) and laboratory hygroscopic growth with a deliquesence point of RH=80 % (Wise et al., 2003) are also shown.**

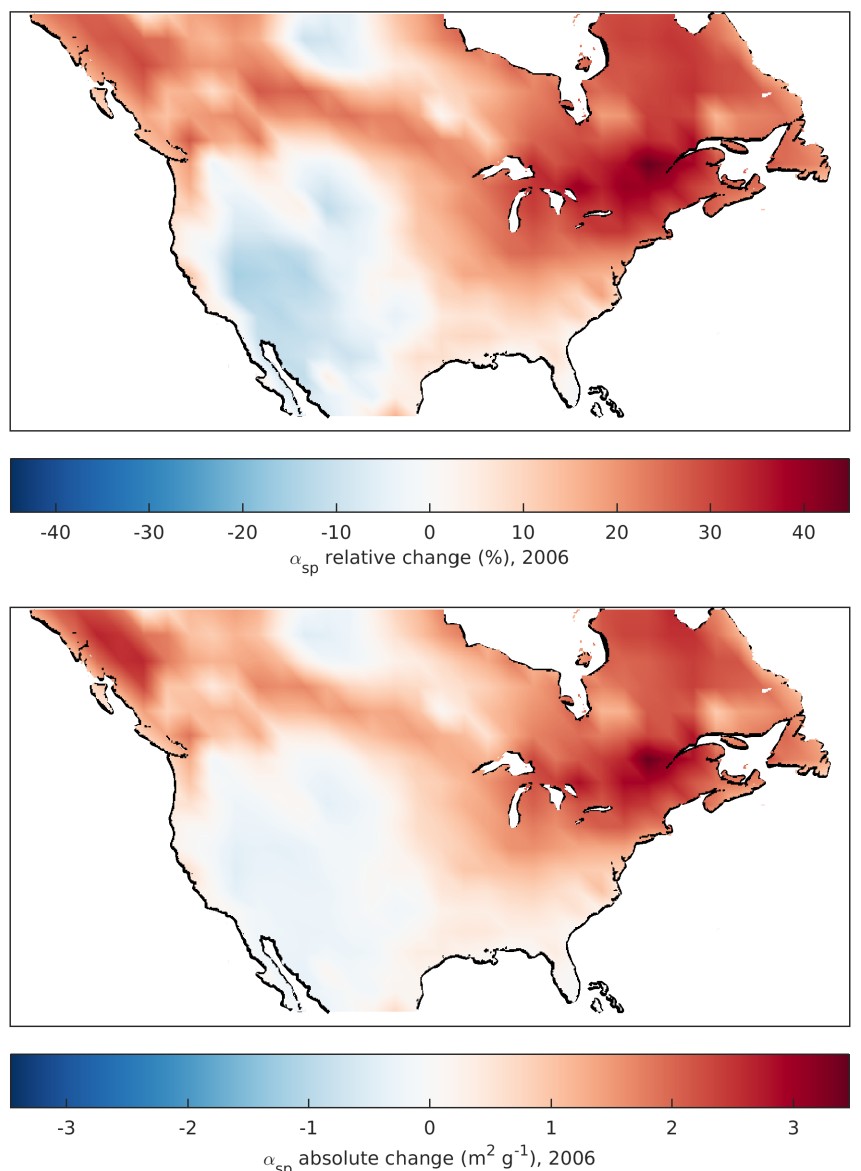

5 **Figure 10: Average relative and absolute change in GEOS-Chem mass scattering efficiency over North America for the year 2006 after implementing revised optical tables for secondary inorganic and organic aerosols.**

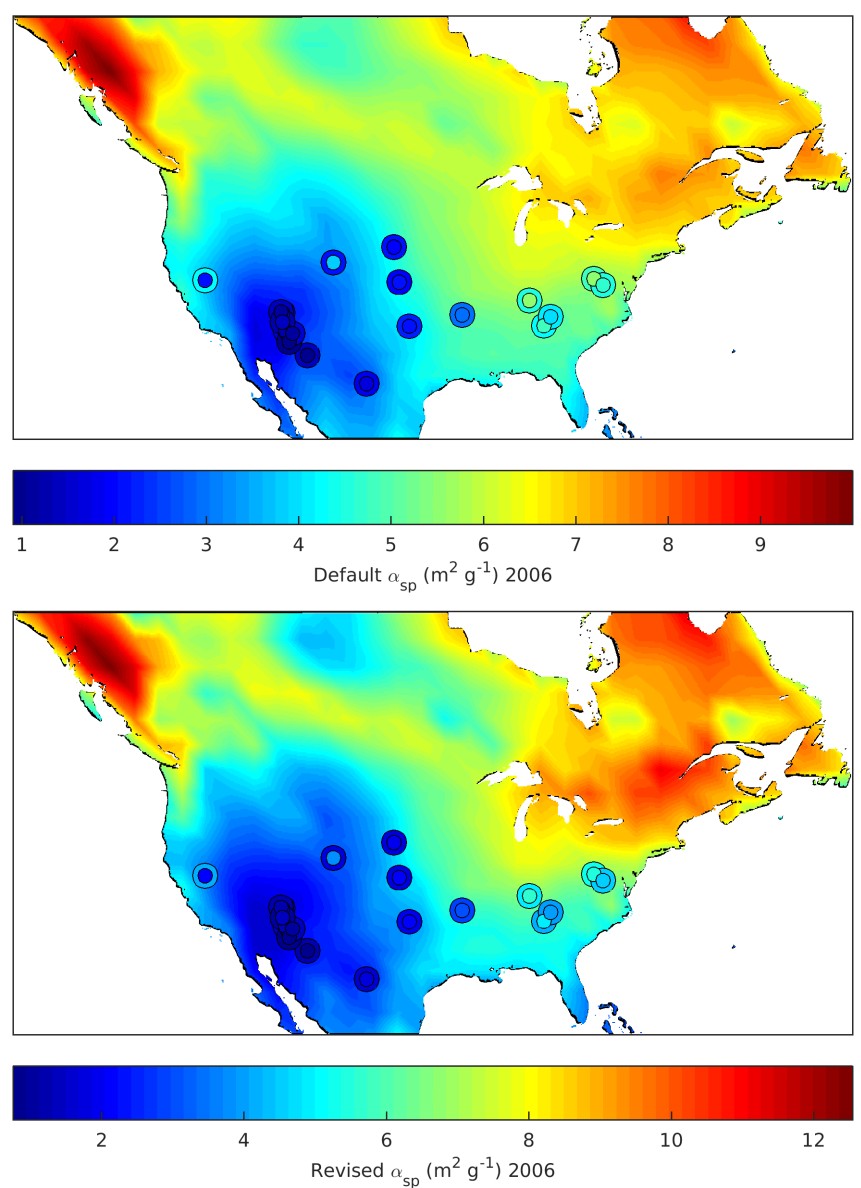

**Figure 11: GEOS-Chem annual average mass scattering efficiency (at 550 nm) for the year 2006 using default and revised size and hygroscopicity for secondary inorganic and organic aerosols. Overlaying inner circles represent annual averages of $\alpha_{sp}$ at IMPROVE network sites for the year 2006. Outer rings represent coincident average simulated $\alpha_{sp}$.**

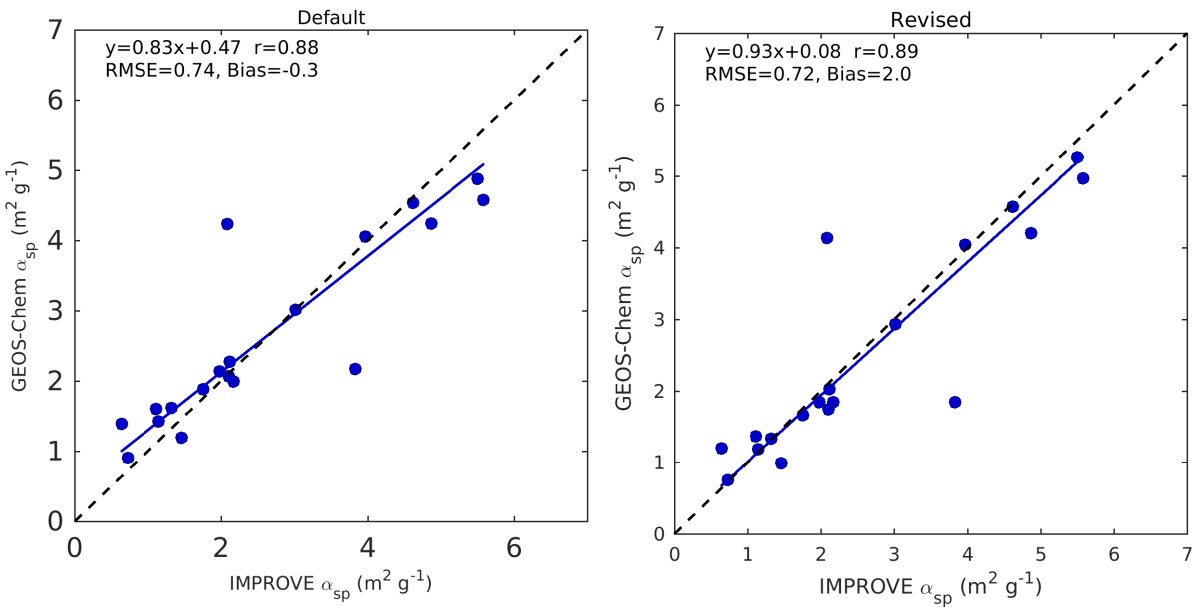

5    **Figure 12: Coincident simulated versus measured average mass scattering efficiency at 550 nm for the year 2006, using default and revised optical tables. Slope, offset and correlation coefficient are inset.**

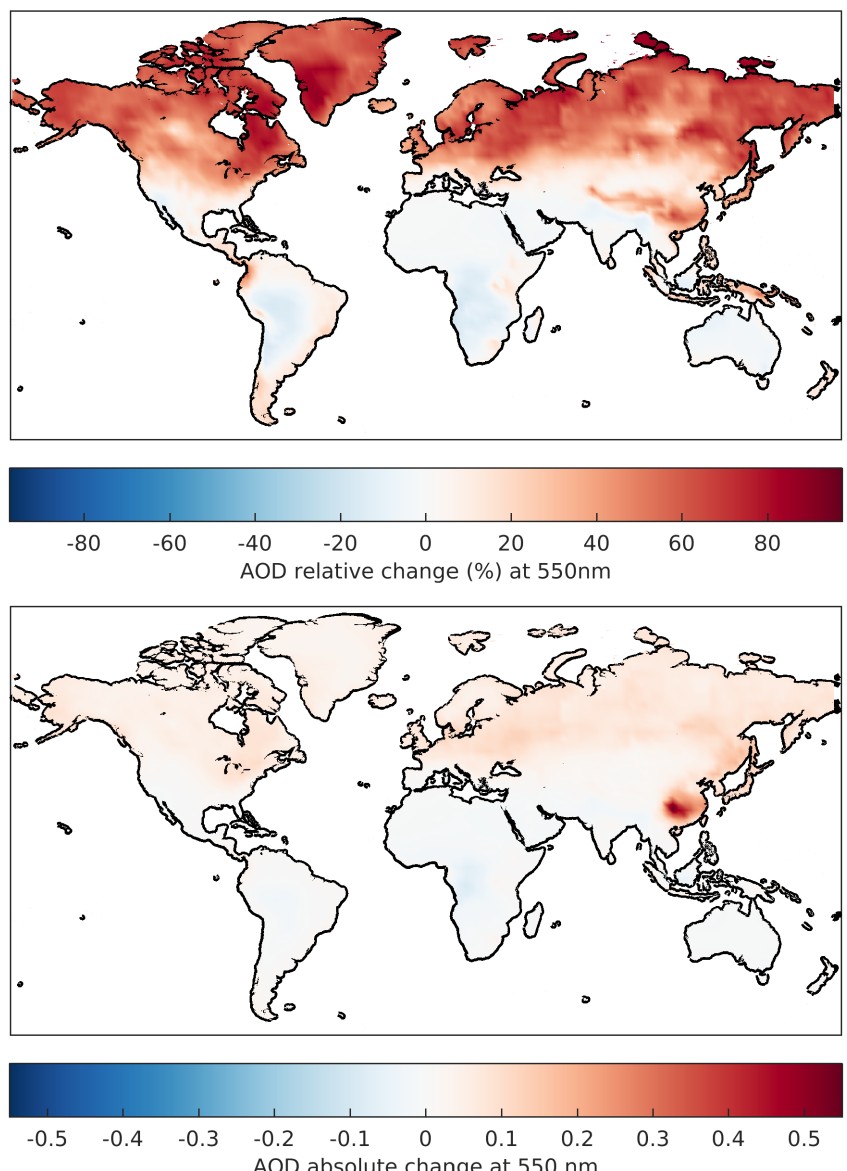

5   **Figure B1: Average relative and absolute change in GEOS-Chem aerosol optical depth at 550 nm globally for the year 2006 after implementing revised optical tables for SIA and OA.**

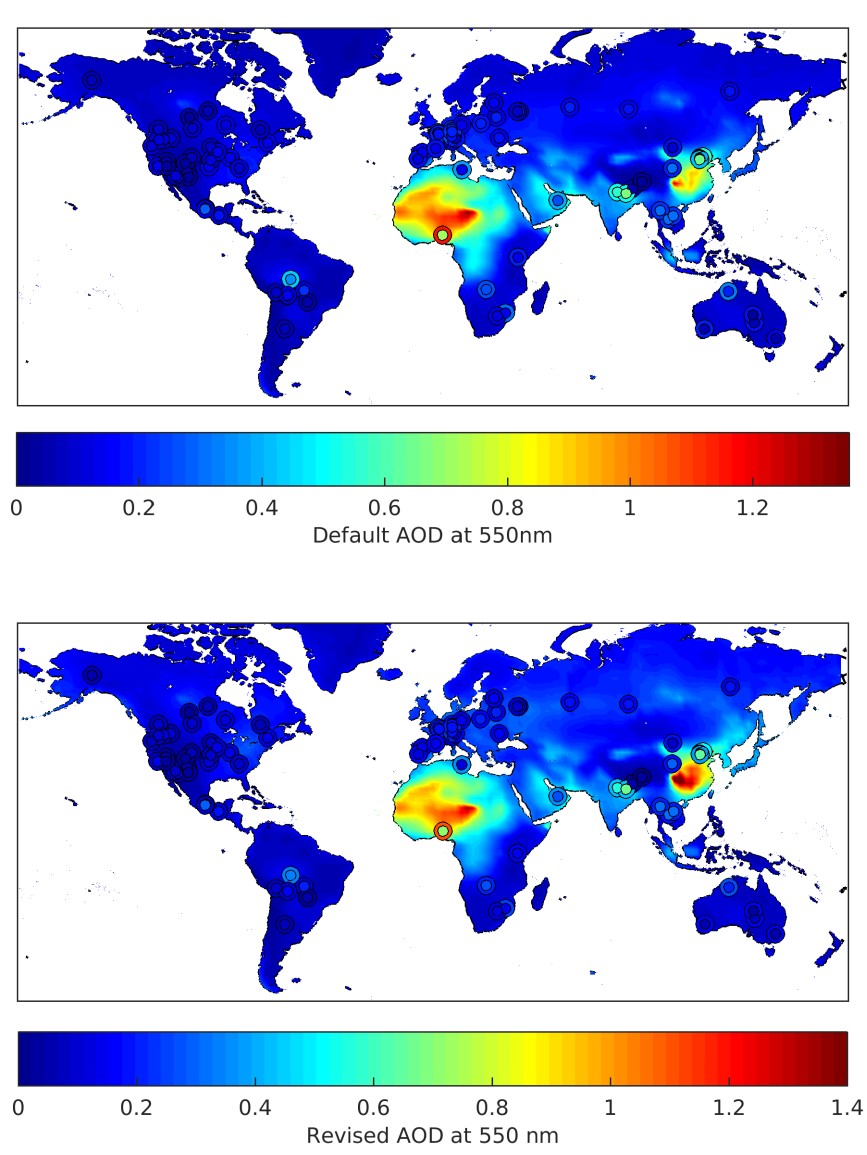

**Figure B2: Global comparison for the year 2006 of AERONET AOD (inner circles) and GEOS-Chem coincident simulated AOD**
5  **(outer rings) using default optical tables.**

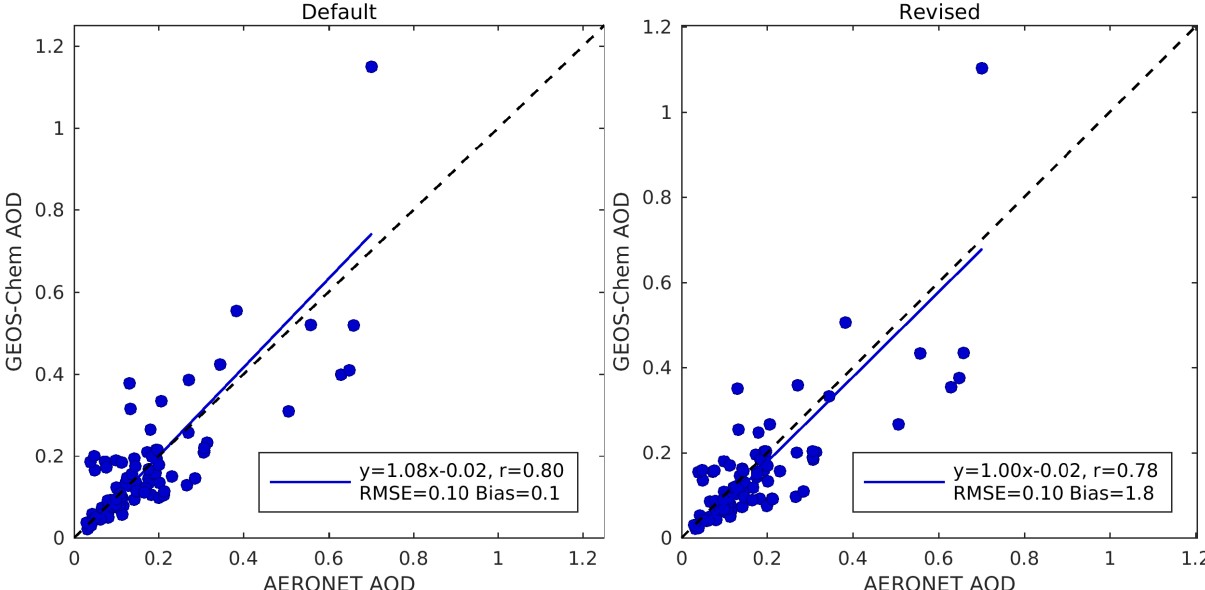

**Figure B.3 Coincident simulated versus measured AOD at 550 nm at AERONET sites for the year 2006, using default and revised sizes and hygroscopicity. Slope, offset and correlation coefficient are inset. The 1:1 line is shown in black.**

| Aerosol | RH | Default | | | | Revised ($\kappa_S$=0.61, $\kappa_O$=0.10) | | | |
|---|---|---|---|---|---|---|---|---|---|
| | | $r_g$ (µm) | $r_{eff}$ (µm) | Q | SSA | $r_g$ (µm) | $r_{eff}$ (µm) | Q | SSA |
| SIA | 0 | 0.069 | 0.121 | 0.902 | 0.965 | 0.058 | 0.101 | 0.603 | 0.959 |
| | 35 | 0.081 | 0.141 | 0.965 | 0.975 | 0.064 | 0.111 | 0.629 | 0.967 |
| | 50 | 0.086 | 0.149 | 0.992 | 0.979 | 0.068 | 0.118 | 0.656 | 0.972 |
| | 70 | 0.093 | 0.163 | 1.062 | 0.983 | 0.078 | 0.135 | 0.742 | 0.981 |
| | 80 | 0.100 | 0.174 | 1.137 | 0.986 | 0.088 | 0.152 | 0.847 | 0.987 |
| | 90 | 0.114 | 0.198 | 1.301 | 0.991 | 0.108 | 0.188 | 1.116 | 0.993 |
| | 95 | 0.131 | 0.227 | 1.517 | 0.994 | 0.135 | 0.234 | 1.500 | 0.997 |
| | 99 | 0.175 | 0.304 | 1.272 | 0.993 | 0.229 | 0.397 | 2.570 | 0.999 |
| OA | 0 | 0.073 | 0.127 | 1.007 | 0.966 | 0.058 | 0.101 | 0.603 | 0.959 |
| | 35 | 0.078 | 0.135 | 0.965 | 0.972 | 0.059 | 0.103 | 0.608 | 0.965 |
| | 50 | 0.080 | 0.139 | 0.947 | 0.975 | 0.060 | 0.104 | 0.610 | 0.963 |
| | 70 | 0.083 | 0.145 | 0.947 | 0.978 | 0.063 | 0.108 | 0.622 | 0.966 |
| | 80 | 0.086 | 0.149 | 0.955 | 0.980 | 0.065 | 0.113 | 0.639 | 0.970 |
| | 90 | 0.092 | 0.159 | 0.990 | 0.984 | 0.073 | 0.125 | 0.696 | 0.977 |
| | 95 | 0.099 | 0.171 | 1.053 | 0.988 | 0.084 | 0.144 | 0.811 | 0.985 |
| | 99 | 0.117 | 0.203 | 1.273 | 0.993 | 0.132 | 0.223 | 1.463 | 0.996 |

**Table A1: Default and revised aerosol size and optical properties for secondary inorganic aerosols (SIA) and organic aerosols (OA) at 550 nm at 8 relative humidity values. Columns indicate geometric mean radius ($r_g$), effective radius ($r_{eff}$), extinction efficiency (Q), and single scattering albedo (SSA). $\kappa_s$ and $\kappa_o$ represent the hygroscopic growth parameters for SIA and OA, respectively.**

| Component | $r_g$ (μm) | σ | $\rho_d$ (g/cm$^3$) |
|---|---|---|---|
| Sulfate/Nitrate/Ammonium | 0.070 | 1.6 | 1.7 |
| Organic Carbon | 0.073 | 1.6 | 1.3 |
| Black Carbon | 0.020 | 1.6 | 1.8 |
| Sea-salt (fine) | 0.085 | 1.5 | 2.2 |
| Sea-salt (coarse) | 0.401 | 1.8 | 2.2 |
| Brown Carbon | 0.073 | 1.6 | 1.3 |
| Dust 1 a-d | 0.030-0.170 | 2.2 | 2.5 |
| Dust 2 | 0.265 | 2.2 | 2.65 |
| Dust 3 | 0.530 | 2.2 | 2.65 |
| Dust 4 | 0.845 | 2.2 | 2.65 |

**Table A2: Current microphysical properties of each aerosol species in GEOS-Chem. $r_g$ represents the dry geometric mean radius (μm) and σ the geometric standard deviation of the lognormal size distributions assumed for each species. $\rho_d$ represents the dry mass densities of each species (g/cm$^3$).**