# Peer review of "Interpretation of Measured Aerosol Mass Scattering Efficiency Over North America Using a Chemical Transport Model"

_Atmospheric Chemistry and Physics, 2018_

## Author Comment (AC1) · 28 Jul 2018

[Figure]

**Figure 3: Average measured versus calculated $\alpha_{sp}$ at IMPROVE sites using GEOS-Chem default and revised optical tables for measurements taken in 0-35 % RH, 35-65 % RH, and 65-95 % RH conditions. The 1:1 line is black. Slope, offset, and correlation coefficient are inset.**

[Figure]

**Figure 4: Average measured versus calculated $\alpha_{sp}$ at IMPROVE sites using GEOS-Chem default and revised optical tables for measurements taken in SIA dominant conditions, OA dominated conditions, Dust dominant conditions, and PM$_{coarse}$ dominated conditions. The 1:1 line is black. Slope, offset, and correlation coefficient are inset.**

[Figure]

**Figure 6: Average measured versus calculated $\alpha_{sp}$ at IMPROVE sites using GEOS-Chem default and revised optical tables for measurements taken in dry conditions (RH<35 %) in winter, spring, summer, and fall. The 1:1 line is black. Slope, offset, and correlation coefficient are inset.**

---

## Referee Comment (RC1) · Anonymous Referee #2 · 5 Aug 2018

The paper studies an important topic for chemistry transport modeling - the mass scattering efficiency that affects the conversion from aerosol mass to aerosol extinction and ultimately AOD. Mass scattering efficiency data from IMPROVE is used to constrain a global CTM (GEOS-Chem, GC). It is found that geometric mean radiuses for both dry secondary inorganic and organic aerosols in original GC configuration need to be decreased, and also the aerosol hygroscopicity curve needs to be adjusted to be less hygroscopic. Overall, the paper should be considered for publication after a major revision (and possibly another review).

General comments 1) there are lots of averaging done in the data analysis. For ex-

ample, the captions for Figures 2, 3, 4, and 6 all start with the word 'average', But, how averaging is done is not described in the method section. In addition, how much is the standard deviation in the average? 2) only R and best fit are shown in all scatter plots. How about RMSE and bias? 3) some discussion of uncertainties here are needed. what are the uncertainties in the measurements of scattering coefficient bsp and surface PM2.5? Likewise, do GEOS fields have any systematic bias in simulating RH?

Specific comments.

P4L10. If IMPROVE PM2.5 is analyzed at 30-50% RH, there will be aerosol water in the aerosol mass measured by IMPROVE. What is the uncertainty here if we assume these are dry particle mass and used in the scattering efficiency calculation? Some consistence is needed or at least discussed between how PM2.5 dry mass is computed in GC vs. how aerosol optics (aerosol hygroscopic growth) are treated in GC. Later in the analysis, the cut off of RH is 35% to be considered as dry (solid) aerosol. Why not 40% as for dry PM2.5?

P4L23. Effectively, b_sp is averaged each day, and then daily averages of b_sp is used together with daily PM2.5 to compute scattering efficiency. How annual average of mass scattering efficiency is computed? How the averages are computed for RH in different ranges or for different compositions (e.g., dust dominated, SIA dominated, etc.)? RH does have a strong diurnal variation.

P6L20-24. The uncertainty of aerosol mass scattering efficiency can also come from the particle composition which essentially affect the particle hygroscopic growth factor, or particle size and density. the scattering efficiency of sulfuric acid vs. ammonium sulfate can be different even for the same size distribution at dry conditions. See "Table 1 in Sensitivity of sulfate direct climate forcing to the hysteresis of particle phase transitions, JGR, 2008". The mass scattering efficiency can also be affected by the mixing state of the particle. The analysis here needs to discuss these uncertainties before

focusing on particle size.

P7L2-3. What is the seasonality of aerosol size distribution?

P7L13-15. Martin et al's paper showed there can be aerosols in solid phase in RH larger than 40%. The phase transition depends on RH history of the particle, not just RH itself. Could this explain the part of overestimation in GC mass scattering efficiency? What is the fraction of solid SIA particles in U.S.? this paper might be helpful – "global distribution of solid and aqueous sulfate aerosols ..., JGR, 2008".

P8L5. What is r_sp used in figure 8?

P11L20. Again, it is the composition that regulates the growth factor and density. The real index of refraction also has the effect on scattering efficiency. In addition, the paper didn't mention effective variance at all. Is it important? References are needed here to support the statements.

P13, L7-8. The aerosol concentration is the same here for both figures. It is likely that the improvement with new optics is within the range of inter-annual variability of alpha_sp itself? It will be more convincing that the validation is done using other years data (say 2007 or 2008).

Figure 5. Caption. Is aerosol effective radius shown for dry (solid) or wet particles?

Figure 9. The GADS assume sulfate as 75% H2SO4, which often has the largest hygroscopic growth. For lab data, please provide reference. Also, lab data clearly shows that ammonium sulfate particles can be in solid phase in range of RH between 40-80%, which may explain why GC has an overestimation in scattering efficiency. How large is the uncertainty in assuming that all SIA are aqueous in RH of 40-80%? If particles are in solid phase in upper troposphere (as models suggest), the large improvement in surface with new optics may not be reflected in the AOD comparison.

Figure 12. add RMSE, and also, how the spectral AOD (or Angstrom exponent) is compared against AERONET? This can be an interesting test for new optics that is a

result of adjusting particle size. Particle size affects both scattering efficiency as well as the spectral AOD slope. Ideally, the new optics should provide an overall improvements for the model.

---

## Referee Comment (RC2) · Anonymous Referee #1 · 6 Aug 2018

This paper examines aerosol mass scattering efficiencies using a chemical transport model and ground-based aerosol composition data. The work is important because aerosol mass scattering efficiencies are required to estimate aerosol radiative properties- both for surface and satellite-derived properties. Understanding model biases as a function of aerosol size and hygroscopic effects is necessary for accurate estimates of aerosol optical properties. The authors 'tune' the model to observations to determine the aerosol size and hygroscopic properties that most closely agree with observations. While the paper is fairly clearly written, it could benefit from clarifications in the method description, as described in the comments below. One of the major weaknesses of the paper is the lack of comparisons of observed and modeled mass

concentrations and light scattering coefficients. Since mass scattering efficiencies are calculated by dividing light scattering by mass, it is important to understand if the model is accurately estimating either of those before attempting to minimize error in size or hygroscopicity. If the model cannot get those parameters correctly, adjusting size or hygroscopicity may be accounting for other errors. Does the model accurately estimate dry conditions? Are the errors in the observations accounted for? I recommend publication after addressing detailed comments below.

Comments

Pg 1:16, Does this reduction correspond to all of North America?

Pg 1:17-20, Are these reductions also reflecting the size shift or just the use of kappa?

Pg 2:4, Perhaps "expansion" is a more accurate term than "growth"?

Pg 2:7, Remove "fine" before "particulate mass concentrations" because this is followed by 10 um and add "PM2.5" before "chemical composition" to be clear that PM10 is not speciated.

Pg2:9, Replace "measurements" with "estimates" because mass scattering efficiency is not measured.

Pg3:8, Define coarse mass as "PM10-PM2.5"

Pg 3:5, bsp isn't measured at all the sites. Perhaps add a line saying that bsp is measured at a subset of sites. This is stated in line 10, but it could be misunderstood here. Perhaps bsp and RH could be left off this list here and then it is clear from line 10 that they are measured at a subset of sites.

Pg3:10, bsp is "reported", not "measured" hourly.

Pg3:14, It is important to consider the impacts of the truncation error for the open air nephelometer and how this can affect scattering where coarse mass is a major contributor.

Pg3:16-17, Can the authors elaborate on why sea salt concentrations are not available before 2005? What parameter is being used to calculate sea salt?

Pg3:20, Do the authors also apply completeness criteria for mass? How many complete years are required for this time period?

Pg3:21, Does average bsp here correspond to ambient RH conditions? Are these the average of daily values or annual values?

Pg4:5, Have the authors considered the influence of this multiplier on mass scattering efficiency? Comparisons of mass and bsp at dry conditions would help show that these assumptions are also appropriate and not affecting efficiencies.

Pg4:10, Please note that the IMPROVE PM2.5 measurement is not a FRM measurement and after 2011 the laboratory RH varied significantly and could contribute a large bias to PM2.5 and PM10 measurements (they are not "dry"). See the data advisory at: http://vista.cira.colostate.edu/improve/Data/QA_QC/Advisory/da0035/da0035_IncreasedRH.pdf

Pg4:11, What RH are these calculated at- model or measured?

Pg4:15, Do the comparisons with the IMPROVE network correspond only to IMPROVE data from 2006 or from an average of 2000-2015? Given the yearly variability, it would more defendable to compare data and modeled values over the same time period.

Pg4: Section 2.3 I have several questions regarding how mass scattering efficiencies are calculated. As mentioned before, truncation errors for the open air nephelometer can impact measured bsp (See Malm and Hand, 2007), essentially removing some (up to half) of the CM scattering – resulting in a discrepancy between scattering and PM10 mass. In addition, dividing ambient bsp by a "dry" PM10 results in an 'enhanced' mass scattering efficiency if bsp and mass don't have the same amount of associated water. PM10 is generally "dry", but as mentioned earlier, after 2011 it can contain a significant amount of water based on the uncontrolled laboratory measurement RH which has nothing to do with ambient RH. If you're comparing data only from 2006

this won't be an issue. But missing the coarse mass contribution to bsp could result in an underestimate of mass scattering efficiency depending on location because in some regions CM is a major contributor to PM10. It is unclear to me why IMPROVE mass is used to estimate the modeled mass scattering efficiency? This would seem to introduce more uncertainty and less consistency in the estimates, especially given the spatial resolution of the model.

It would be helpful to see a comparison of modeled and measured mass concentrations and bsp. Do the predictions of SIA and OA generally agree with the observations? Can the model accurately predict measured bsp? It seems it would be important to know the model performance of the parameters that go into estimating mass scattering efficiency before trying to tune the model results using size and hygroscopicity. While perhaps the new revised estimates are more accurate, it is hard to know whether they are just accounting for other issues within the model or the measurements. Without the basic comparisons, the comparison is less convincing.

Pg5:4, What is "v" as the kappa subscript refer to?

Pg5:20, Include "annual" before "average".

Pg5:22, It would help to at least mention that the figure also includes revised estimates that will be introduced later, otherwise the figures appear to be referred to out of numeric order (figure 3 is discussed before figure 2, right side- etc.).

Pg6:1, Is the PM2.5 here referring to observations or model estimates?

Pg6:1, Add "fine" before "dust"

Pg6:12, Can the authors reconcile studies that demonstrate that even at low RH particles will still have water associated with them? (e.g., Santarpia et al., 2004, 2005; Carrico et al., 2000, 2003, 2005; ten Brink et al., 2006; Malm and Day, 2001; Malm et al., 2003, 2005; Xu et al., 2002; Im et al., 2001; Eldering et al., 2002)

Pg8:20, The authors have seemingly tuned their model to the parameters that give the

closest agreement with observations so it isn't surprising that the revised estimates agree more closely. However, they don't comment on why the model inherently had too large of sizes. Have other studies also observed this? Is it consistent with size distributions measured at surface sites? Given some of the issues with observations mentioned earlier, it would be important to understand whether the model adjustments are just accounting for some of the measurement biases. This could be partly answered if comparisons of modeled and measured mass concentrations and bsp were performed.

Pg9:18, Mass scattering efficiencies of 10-12 m2/g are exceptionally high unless they correspond to enhanced scattering efficiencies with higher RH bsp and dry mass. Providing the RH that these values correspond to is important.

Pg11:14, Again it would be helpful to show a comparison of measured and modeled bsp and of mass separately.

Pg12:4, Again, see the earlier advisory. IMPROVE PM2.5 and PM10 measurements are not FRM. Pg12:8, Why is it necessary to partition IMPROVE OM? Unless I have missed something, hygroscopic calculations for IMPROVE mass weren't done as part of this paper?

Figures Pg19:4, Figure 1, Provide the wavelength and RH (or state ambient) for bsp. The sentence starting with "overlaying circles" seems redundant.

Pg 20, Figure 2, please include the years included in the comparison, and the wavelength corresponding to the efficiency. It would help to also denote the parts of the figure as well.

Pg20, Figure 3: Similar comment regarding wavelength as other figures (this is helpful for readers if they are scanning figures and don't have to dig through the text for this information). Please include the years this comparison was made over. Also comment on "default" and "revised".

Pg21, Figure 4: Include wavelength. Define SIA, OA, PMcoarse. Are these delin-eations based on modeled or measured composition measurements? Please state in caption. Also include RH ranges (ambient?) and years of comparison.

Pg 21, Figure 5: Please include wavelength. What kappa was used for OC and SIA?

Pg 22, Figure 6: Please include wavelength and years of comparison.

Pg 23, Figure 8: Please define kappa,s and kappa,o.

Pg23, Figure 9: Please define SIA and OA. "Hygroscopic" has a typo.

Pg24, Figure 10: Define SIA and OA.

Pg25, Figure 11: Please provide wavelength and RH (ambient?), define SIA and OA. This color table makes it hard to see the sites in the Southwest.

Pg26: Figure 12: Please provide wavelength and RH (ambient?)

Pg 28, Figure B2: This color table makes it hard to see the comparisons.

Pg 30, Table A1, Define SIA and OA, kappa,s and kappa,o (use similar subscripts as SIA and OA?)

References: Carrico, C. M., M. J. Rood, J. A. Ogren, C. Neusüß, A. Wiedensohler, and J. Heintzenberg, Aerosol optical properties at Sagres, Portugal during ACE-2, Tellus, 52B, 694-715, 2000.

Carrico, C. M., M. H. Bergin, J. Xu, K. Baumann, and H. Maring, Urban aerosol radiative properties: Measurements during the 1999 Atlanta Supersite Experiment, J. Geophys. Res., 108(D7), 8422, doi:10.1029/2001JD001222, 2003.

Carrico, C. M., S. M. Kreidenweis, W. C. Malm, D. E. Day, T. Lee, J. Carrillo, G. R. McMeeking, and J. L. Collett, Jr., Hygroscopic growth behavior of a carbon-dominated aerosol in Yosemite National Park, Atmos. Environ., 39, 1393-1404, 2005.

Eldering, A., J. A. Ogren, Z. Chowdhury, L. S. Hughes, and G. R. Cass, Aerosol optical

properties during INDOEX based on measured aerosol particle size and composition, J. Geophys. Res.,107(D22), 8001, doi:10.1029/2001JD001572, 2002.

Im, J.-S., V. K. Saxena, and B. N. Wenny, An assessment of hygroscopic growth factors for aerosols in the surface boundary layer for computing direct radiative forcing, J. Geophys. Res., 106(D17), 20213-20224, 2001

Malm W.C. and D. E. Day, Estimates of aerosol species scattering characteristics as a function of relative humidity, Atmos. Environ., 35, 2845-2860, 2001.

Malm W.C., D. E. Day, S. M. Kreidenweis, J. L. Collett, Jr., T. Lee, T., Humidity dependent optical properties of fine particles during the Big Bend Regional Aerosol and Visibility Observational study (BRAVO), J. Geophys. Res. 108(D9), 4279, doi:10.1029/2002JD002998, 2003.

Malm, W. C., D. E. Day, S. M. Kreidenweis, J. L. Collett, Jr., C. Carrico, G. McMeeking, and T. Lee, Hygroscopic properties of organic-laden aerosol, Atmos. Environ., 39, 4969-4982, 2005.

Santarpia, J. L., R. Gasparini, R. Li, and D. R. Collins, Diurnal variations in the hygroscopic growth cycles of ambient aerosol populations, J. Geophys. Res.,110, D03206, doi:10.1029/2004JD005279, 2005.

ten Brink, H. M., J. P. Veefkind, A. Waijers-Ijpelaan, and J. C. Van der Hage, Aerosol light scattering in the Netherlands, Atmos. Environ., 30(24), 4251-4261, 1996.

Xu, J., M. H. Bergin, X. Yu, G. Liu, J. Zhao, C. M. Carrico, and K. Baumann, Measurement of aerosol chemical, physical and radiative properties in the Yangtze delta region of China, Atmos. Environ., 36, 161-173, 2002.

---

## Author Comment (AC2) · 29 Oct 2018

**Reviewer: 1**

*Comments:*

This paper examines aerosol mass scattering efficiencies using a chemical transport model and ground-based aerosol composition data. The work is important because aerosol mass scattering efficiencies are required to estimate aerosol radiative properties- both for surface and satellite-derived properties. Understanding model biases as a function of aerosol
10 size and hygroscopic effects is necessary for accurate estimates of aerosol optical properties. The authors 'tune' the model to observations to determine the aerosol size and hygroscopic properties that most closely agree with observations. While the paper is fairly clearly written, it could benefit from clarifications in the method description, as described in the comments below. One of the major weaknesses of the paper is the lack of comparisons of observed and modeled mass concentrations and light scattering coefficients. Since mass scattering efficiencies are calculated by dividing light scattering by mass, it is
15 important to understand if the model is accurately estimating either of those before attempting to minimize error in size or hygroscopicity. If the model cannot get those parameters correctly, adjusting size or hygroscopicity may be accounting for other errors. Does the model accurately estimate dry conditions? Are the errors in the observations accounted for? I recommend publication after addressing detailed comments below.

Thank you for the positive comments. There seems to be a misunderstanding regarding our methods. The purpose of the
20 manuscript is to evaluate mass scattering efficiency since that is largely assumed in GEOS-Chem. Modeled mass scattering efficiency is not calculated from modeled mass and $b_{sp}$, so evaluation of modeled mass and $b_{sp}$ is a separate question. Rather than using modeled light scattering and mass to calculate mass scattering efficiencies, we calculate mass scattering efficiency using IMPROVE speciation measurements alongside aerosol physical and optical properties from the model. This removes the influence of possible biases in model mass or composition from the analysis. Thus the comparison of modeled
25 and measured mass concentrations and $b_{sp}$ are irrelevant. The purpose of the manuscript is to evaluate mass scattering efficiency since that is largely assumed in GEOS-Chem. Modeled mass scattering efficiency is not calculated from modeled mass and $b_{sp}$, so evaluation of modeled mass and $b_{sp}$ is a separate question. We clarified this in the manuscript:

Page 4 Line 8-12: "The majority of our analysis focuses on the accuracy of the GEOS-Chem parameterization of mass
30 scattering efficiency based on optical parameters given in Table A1. These default aerosol physical and optical properties are defined by the Global Aerosol Data Set (GADS) (Koepke et al., 1997), as implemented by Martin et al. (2003), with modifications to dry size distributions (Drury et al., 2010) and dust mass partitioning (Ridley et al., 2012). After evaluating and improving this parameterization, implications are examined using the full GEOS-Chem simulation in section 3.3."

35 Further clarifications in the methods are provided below.

*Specific Comments:*

5  Pg 1:16, Does this reduction correspond to all of North America?

Errors in observations are accounted for as discussed in detail below.

The overall bias is reduced from 82% to 9% in dry conditions. This takes into account all sites in North America. The words "overall" and "at IMPROVE sites" have been added into this sentence to clarify:

Page 1 Line 14-17: "Inhibiting hygroscopic growth of SIA below 35% RH and decreasing the dry geometric mean radius, from 0.069 μm for SIA and 0.073 μm for OA to 0.058 μm for both aerosol types, significantly decreased the overall bias observed at IMPROVE sites in dry conditions from 82% to 9%."

15  Pg 1:17-20, Are these reductions also reflecting the size shift or just the use of kappa?

Clarification has been added to indicate that these changes reflect both the size shift and the use of kappa.

Page 1 Line 19-22: "Incorporating these changes in aerosol size and hygroscopicity into the GEOS-Chem model resulted in

20  an increase of 16% in simulated average $\alpha_{sp}$ over North America, with larger increases of 25% to 45% in northern regions with high RH and hygroscopic aerosol fractions, and decreases in $\alpha_{sp}$ up to 15% in the southwestern U.S. where RH is low."

Pg 2:4, Perhaps "expansion" is a more accurate term than "growth"?

25  "growth" has been changed to "expansion" (now Page 2 Line 9)

Pg 2:7, Remove "fine" before "particulate mass concentrations" because this is followed by 10 um and add "PM2.5" before "chemical composition" to be clear that PM10 is not speciated.

Thank you, these changes have been made.

30  Page 2 Line 10-13 "The Interagency Monitoring of Protected Visual Environments (IMPROVE) network offers long-term collocated measurements since 1987 of particle scatter ($b_{sp}$), relative humidity (RH), particulate mass concentrations less than 10 μm ($PM_{10}$) and less than 2.5 μm ($PM_{2.5}$) as well as $PM_{2.5}$ chemical composition at sites across the United States and Canada (Malm et al., 1994; Malm et al., 2004).

Pg 2:9, Replace "measurements" with "estimates" because mass scattering efficiency is not measured.

Page 2 Line 14: "measurements" has been replaced with "estimates".

Pg 3:8, Define coarse mass as "PM10-PM2.5"

This definition has been added.

5 Page 3 Line 9-10: "Collected $PM_{10}$ undergoes gravimetric analysis for total particulate mass less than 10μm, allowing for the determination of coarse mass ($PM_{10}$-$PM_{2.5}$)"

Pg 3:5, bsp isn't measured at all the sites. Perhaps add a line saying that bsp is measured at a subset of sites. This is stated in line 10, but it could be misunderstood here. Perhaps bsp and RH could be left off this list here and then it is clear from line 10 that they are measured at a subset of sites.

10 Clarification has been added to indicate that bsp and RH are measured at a subset of IMPROVE sites.

Page 3 Line 4:6: "The network offers measurements of $PM_{2.5}$ speciation, $PM_{2.5}$ and $PM_{10}$ gravimetric mass, and collocated measurements of $b_{sp}$ and RH at a subset of sites that we interpret to understand mass scattering efficiency."

Pg 3:10, bsp is "reported", not "measured" hourly.

15 Thank you, this has been corrected (now page 3, line 12).

Pg 3:14, It is important to consider the impacts of the truncation error for the open air nephelometer and how this can affect scattering where coarse mass is a major contributor.

To investigate the possible bias introduced by nephelometer truncation of coarse mass, the fourth panel of Figure 4 shows calculated vs. measured mass scattering efficiency in conditions dominated by coarse mass ($PM_{coarse}/PM_{10} > 0.6$). While

20 there is a slight overestimation of mass scattering efficiency in calculated $\alpha_{sp}$ compared to measured $\alpha_{sp}$ in $PM_{coarse}$ dominant conditions, this overestimation is more pronounced in conditions dominated by secondary inorganic and organic aerosols. It appears that on average, the bias introduced by nephelometer truncation is not as significant as the bias in the representation of SIA and OA parameters. For this reason, we focus on the representation of SIA and OA parameters.

Page 5 Line 20-22: "Although the OPTEC open air nephelometer reduces truncation error compared with other

25 nephelometers, truncation error can be significant for coarse particles (Hand and Malm, 2007; Lowenthal and Kumar, 2006). Thus our analysis below focuses on conditions dominated by fine mode aerosols, and mechanisms affecting fine mode aerosols."

Pg 3:16-17, Can the authors elaborate on why sea salt concentrations are not available before 2005? What parameter is being

30 used to calculate sea salt?

IMPROVE uses the chloride ion to estimate sea salt concentrations by SS=1.8[Cl⁻]. From 2000-2003, many samples at interior sites reported negative chloride concentrations. This has been associated with chloride backgrounds present on the Nylon filters used during this time period (see data advisory: http://vista.cira.colostate.edu/improve/Data/QA_QC/Advisory/da0036/da0036_chloride_loss.pdf) At the time IMPROVE data was downloaded, SS concentrations were unavailable from 2000 through 2004.

The following sentence has been added to clarify that sea-salt is not reported:

Page 3 Line 25-27: Sea salt aerosols are excluded from the analysis from 2000-2004, as reliable estimates of sea salt concentrations were unavailable during this period. We exclude coastal sites during this period, as sea salt can contribute significantly to $b_{sp}$ in coastal conditions of high RH due to its highly hygroscopic nature (Lowenthal and Kumar, 2006).

Pg 3:20, Do the authors also apply completeness criteria for mass? How many complete years are required for this time period?

Clarification has been added regarding completeness criteria for mass. This section now reads "We use only days with coincident mass and scatter measurements, and a minimum of 23 hourly measurements per day, to reduce influence of meteorological interference. Additionally, only sites with a minimum of 90 days of measurements are included in the analysis."

Pg 3:21, Does average bsp here correspond to ambient RH conditions? Are these the average of daily values or annual values?

Average $b_{sp}$ corresponds to ambient RH conditions, and these are the average of hourly $b_{sp}$ values. The following clarification has been added:

Page 3 Line 31-32: "Figure 1 shows at the 28 sites used in this study the average hourly $b_{sp}$ at ambient RH and the average 24h $PM_{10}$ and $PM_{2.5}$ measured between 2000-2010."

Pg 4:5, Have the authors considered the influence of this multiplier on mass scattering efficiency? Comparisons of mass and bsp at dry conditions would help show that these assumptions are also appropriate and not affecting efficiencies.

Interesting idea. Although this multiplier is one of several parameters that influence modelled mass scattering efficiency in section 3.3, it does not affect the bulk of our analysis since the entirety of our analysis prior to section 3.3 does not use model output, as is explained in more detail in response to a later comment. Thus we focus on size and hygroscopicity which more directly influence mass scattering efficiency.

Unfortunately, we are unable to compare modelled and measured mass and $b_{sp}$ in dry conditions, as IMPROVE measures $b_{sp}$ in ambient conditions, and you mention below, the RH at which $PM_{10}$ mass is measured is not dry.

Pg 4:10, Please note that the IMPROVE PM2.5 measurement is not a FRM measurement and after 2011 the laboratory RH varied significantly and could contribute a large bias to PM2.5 and PM10 measurements (they are not "dry"). See the data advisory at: http://vista.cira.colostate.edu/improve/Data/QA_QC/Advisory/da0035/da0035_IncreasedRH.pdf

Thank you for bringing this to our attention. Data from 2011-2015 has been removed from the analysis to address this concern. The removal of these four years of data, and the implementation of more rigorous completeness criteria for mass has resulted in slight changes in most of our plots. Most of our results have remained the same, however $\kappa_s$ has increased from 0.58 to 0.61.

5  It should be noted that in this analysis, $PM_{2.5}$ and $PM_{10}$ measurements are not assumed to be dry. It is assumed that PM2.5 and $PM_{10}$ mass measurements are performed at 40% RH. Further clarification on this has been added in section 2.3.

Section 2.3 (Page 5 Line 10-12): "We define $\alpha_{sp}$ operationally here based on optical measurements at ambient RH, and PM measurements at controlled RH (treated as 40% RH for consistency with IMPROVE protocols prior to 2011). At 40% RH, hygroscopic components of $PM_{10}$ will have associated water, and thus measured $PM_{10}$ mass is not treated as dry."

10  Pg 4:11, What RH are these calculated at- model or measured?

Particle scatter and aerosol optical depth are calculated at modelled ambient RH. The word "modelled" has been added to this sentence for clarification.

Page 4 Line 20-22: "Particle scatter and aerosol optical depth are calculated at modelled ambient RH based on dry species mass concentrations and aerosol physical and optical properties."

15  Pg 4:15, Do the comparisons with the IMPROVE network correspond only to IMPROVE data from 2006 or from an average of 2000-2015? Given the yearly variability, it would more defendable to compare data and modeled values over the same time period.

Only simultaneous hourly modelled and measured mass scattering efficiencies are used in the comparison. "Coincident"  and "over the same time period" have been added to the sentence for clarification:

20  Page 4 Line 26-28 "We simulate $PM_{10,}$ $PM_{2.5}$ and $b_{sp}$, allowing for the comparison of model mass scattering efficiency coincident to that measured at IMPROVE network sites over the same time period over North America."

Pg 4: Section 2.3 I have several questions regarding how mass scattering efficiencies are calculated. As mentioned before, truncation errors for the open air nephelometer can impact measured bsp (See Malm and Hand, 2007), essentially removing some (up to half) of the CM scattering – resulting in a discrepancy between scattering and PM10 mass.

25  To investigate the possible bias introduced by nephelometer truncation of coarse mass, the fourth panel of Figure 4 shows calculated vs. measured mass scattering efficiency in conditions dominated by coarse mass ($PM_{coarse}/PM_{10} > 0.6$). While there is a slight overestimation of mass scattering efficiency in calculated $\alpha_{sp}$ compared to measured $\alpha_{sp}$ in $PM_{coarse}$ dominant conditions, this overestimation is more pronounced in conditions dominated by secondary inorganic and organic aerosols. It appears that on average, the bias introduced by nephelometer truncation is not as significant as the bias in the representation
30  of SIA and OA parameters. For this reason, we focus on the representation of SIA and OA parameters.

Page 5 Line 20-22"Although the OPTEC open air nephelometer reduces truncation error compared with other nephelometers, truncation error can be significant for coarse particles (Hand and Malm, 2007; Lowenthal and Kumar, 2006).

Thus our analysis below focuses on conditions dominated by fine mode aerosols, and mechanisms affecting fine mode aerosols."

In addition, dividing ambient bsp by a "dry" PM10 results in an 'enhanced' mass scattering efficiency if bsp and mass don't

5 have the same amount of associated water. PM10 is generally "dry", but as mentioned earlier, after 2011 it can contain a significant amount of water based on the uncontrolled laboratory measurement RH which has nothing to do with ambient RH. If you're comparing data only from 2006 this won't be an issue. But missing the coarse mass contribution to bsp could result in an underestimate of mass scattering efficiency depending on location because in some regions CM is a major contributor to PM10.

10 In section 2.3, we acknowledge that multiple definitions of mass scattering efficiency exist, and given the nature of the measurements available, we define mass scattering efficiency for this analysis as $b_{sp}$ at ambient RH divided by gravimetric $PM_{10}$ mass at 40% RH.

The following sentence has been added to clarify that gravimetric $PM_{10}$ mass is not treated as dry. Page 5, Line 11-12: "At 40% RH, hygroscopic components of $PM_{10}$ will have associated water, and thus measured $PM_{10}$ mass is not treated as dry."

15 We also exclude data after 2010 when lab RH becomes variable. Page 3, Line 24-25: "We exclude data after 2010 to address concerns about variable laboratory RH for $PM_{10}$ measurement after 2010."

It is unclear to me why IMPROVE mass is used to estimate the modeled mass scattering efficiency? This would seem to introduce more uncertainty and less consistency in the estimates, especially given the spatial resolution of the model.

This approach enables isolation of the mass scattering efficiency used in GEOS-Chem from the species concentrations. We

20 added text to clarify the method.

Page 4 line 8-12: "The majority of our analysis focuses on the accuracy of the GEOS-Chem parameterization of mass scattering efficiency based on optical parameters given in Table A1. These default aerosol physical and optical properties are defined by the Global Aerosol Data Set (GADS) (Koepke et al., 1997), as implemented by Martin et al. (2003), with

25 modifications to dry size distributions (Drury et al., 2010) and dust mass partitioning (Ridley et al., 2012). After evaluating and improving this parameterization, implications are examined using the full GEOS-Chem simulation in section 3.3."

It would be helpful to see a comparison of modeled and measured mass concentrations and bsp. Do the predictions of SIA and OA generally agree with the observations? Can the model accurately predict measured bsp? It seems it would be

30 important to know the model performance of the parameters that go into estimating mass scattering efficiency before trying to tune the model results using size and hygroscopicity. While perhaps the new revised estimates are more accurate, it is hard to know whether they are just accounting for other issues within the model or the measurements. Without the basic comparisons, the comparison is less convincing.

There seems to be a misunderstanding here regarding our methods. Rather than using model output of mass scattering efficiency to refine size and hygroscopicity parameters, we calculate mass scattering efficiency using IMPROVE speciation measurements alongside aerosol physical and optical properties from the model. We then use this calculated mass scattering efficiency to refine size and hygroscopicity parameters. This removes the influence of possible biases in model output from the analysis. Thus the comparison of modeled and measured mass concentrations and $b_{sp}$ are irrelevant.

On page 4, line 8-12 we added "The majority of our analysis focuses on the accuracy of the GEOS-Chem parameterization of mass scattering efficiency based on optical parameters given in Table A1. These default aerosol physical and optical properties are defined by the Global Aerosol Data Set (GADS) (Koepke et al., 1997), as implemented by Martin et al. (2003), with modifications to dry size distributions (Drury et al., 2010) and dust mass partitioning (Ridley et al., 2012). After evaluating and improving this parameterization, implications are examined using the full GEOS-Chem simulation in section 3.3."

GEOS-Chem model output is used in this analysis in section 3.3 only, when we investigate how the changes to size and hygroscopicity impact model output.

Pg 5:4, What is "v" as the kappa subscript refer to?

Thanks for bringing this to our attention, the subscript "v" has been removed from equation 5 (formerly equation 4) and page 6, line 7.

Pg 5:20, Include "annual" before "average".

The average mass scattering efficiencies described here are averages over the entire sampling period at each IMPROVE site, not annual averages. A sentence has been added to section 2.3 describing how average mass scattering efficiency is calculated for each site.

Page 5 Line16-19: "To reduce the impacts of meteorological variation on the comparison of measured and calculated mass scattering efficiency, we perform averages of hourly $b_{sp,calc}$, $b_{sp,meas}$, and $PM_{10}$ over the entire sampling period at each IMPROVE site $i$. Eq. (3) is then used to obtain average calculated and measured mass scattering efficiency at each site.

$$\alpha_{sp,avg,i} = \frac{b_{sp,avg,i}}{PM_{10,avg,i}} \qquad\qquad (3)"$$

Page 6 Line 12-13: "Each point represents the average $\alpha_{sp}$ over the entire sampling period at each IMPROVE site."

Pg 5:22, It would help to at least mention that the figure also includes revised estimates that will be introduced later, otherwise the figures appear to be referred to out of numeric order (figure 3 is discussed before figure 2, right side- etc.).

This has been added to page 6 line 15: "(The right panel of Figure 2 is discussed below.)"

Pg 6:1, Is the PM2.5 here referring to observations or model estimates?

The $PM_{2.5}$ is referring to observations. The word "measured" has been added before $PM_{2.5}$ to clarify.

Page 6 Line 26-28: Figure 4 shows in blue average measured vs calculated $\alpha_{sp}$ using default optical tables for conditions where measured $PM_{2.5}$ is dominated (>60%) by secondary inorganic aerosol, organic aerosol and fine dust, as well conditions where $PM_{10}$ is dominated (>60%) by $PM_{coarse}$ ($PM_{10}$-$PM_{2.5}$).

Pg 6:1, Add "fine" before "dust"

This change has been made. The sentence now reads:

Page 6 Line 26-28: Figure 4 shows in blue average measured vs calculated $\alpha_{sp}$ using default optical tables for conditions where measured $PM_{2.5}$ is dominated (>60%) by secondary inorganic aerosol, organic aerosol and fine dust, as well conditions where $PM_{10}$ is dominated (>60%) by $PM_{coarse}$ ($PM_{10}$-$PM_{2.5}$)

Pg 6:12, Can the authors reconcile studies that demonstrate that even at low RH particles will still have water associated with them? (e.g., Santarpia et al., 2004, 2005; Carrico et al., 2000, 2003, 2005; ten Brink et al., 2006; Malm and Day, 2001; Malm et al., 2003, 2005; Xu et al., 2002; Im et al., 2001; Eldering et al., 2002).

While we later inhibit hygroscopic growth of SIA below 35% RH, we reconcile this by allowing hygroscopic growth of organics at low RH.

Pg 8:20, The authors have seemingly tuned their model to the parameters that give the closest agreement with observations so it isn't surprising that the revised estimates agree more closely. However, they don't comment on why the model inherently had too large of sizes. Have other studies also observed this? Is it consistent with size distributions measured at surface sites? Given some of the issues with observations mentioned earlier, it would be important to understand whether the model adjustments are just accounting for some of the measurement biases. This could be partly answered if comparisons of modeled and measured mass concentrations and bsp were performed.

The initial aerosol sizes used in the current version of GEOS-Chem were based on summertime aerosol measurements (Drury et al., 2011), when aerosols are larger. This serves as a reasonable explanation for why the model inherently has too large of sizes in seasons other than summer. The following sentence has been added for clarification:

Page 8 Line 28-29: "This annual radius is smaller than the GEOS-Chem default sizes of SIA and OA that were informed by summertime measurements alone (Drury et al., 2010)."

Available aerosol size distribution measurements at surface sites in North America are mostly limited to small particle size ranges, and therefore do not include particles large enough to be useful for comparisons in our study. We have however

included the following sentence to compare our seasonal aerosol size results to measurements performed by Levin et al. in 2006:

Page 8 Line 26-27: "The spring and summer radii are consistent with accumulation mode size distribution measurements performed by Levin et al. (2009) in the spring and summer of 2006."

As previously stated, modelled mass concentrations have not been used in this analysis and are thus irrelevant for this discussion.

Pg 9:18, Mass scattering efficiencies of 10-12 m2/g are exceptionally high unless they correspond to enhanced scattering efficiencies with higher RH bsp and dry mass. Providing the RH that these values correspond to is important.

As the definition of mass scattering efficiency used in this study is $b_{sp\ (ambient\ RH)}$ / $PM_{10\ (40\%\ RH)}$, mass scattering efficiencies will be enhanced in conditions of high humidity. The following sentence has been added:

Page 10 Line 22-23: "The elevated mass scattering efficiencies in the northwest can be attributed in part to the high average RH in this region of 83%."

Pg 11:14, Again it would be helpful to show a comparison of measured and modeled bsp and of mass separately.

Again, the GEOS-Chem mass scattering efficiency parameterization is not calculated from the ratio of modelled mass to $b_{sp}$. We have added the following text to clarify.

Page 4 Line 8-12: "The majority of our analysis focuses on the accuracy of the GEOS-Chem parameterization of mass scattering efficiency based on optical parameters given in Table A1. These default aerosol physical and optical properties are defined by the Global Aerosol Data Set (GADS) (Koepke et al., 1997), as implemented by Martin et al. (2003), with modifications to dry size distributions (Drury et al., 2010) and dust mass partitioning (Ridley et al., 2012). After evaluating and improving this parameterization, implications are examined using the full GEOS-Chem simulation in section 3.3."

Page 4 Line 22-24: "The GEOS-Chem aerosol simulation has been extensively evaluated with observations of mass (van Donkelaar et al., 2015; Li et al., 2016), composition (Kim et al., 2015; Marais et al., 2016; Philip et al., 2014a), and scatter (Drury et al., 2010)."

Pg 12:4, Again, see the earlier advisory. IMPROVE PM2.5 and PM10 measurements are not FRM.

Thank you, this has been addressed by removing data from 2011-2015 from the analysis.

Pg 12:8, Why is it necessary to partition IMPROVE OM? Unless I have missed something, hygroscopic calculations for IMPROVE mass weren't done as part of this paper?

In order to calculate $\alpha_{sp}$ from IMPROVE speciation measurements using Eq. A2:

$$\alpha_{sp} = \frac{B_{sp}}{PM_{10}} = \frac{\sum_{species,i} \frac{\frac{3}{4}\left(\frac{R_{w,i}}{R_{d,i}}\right)^2 * \frac{M_{d,i}}{PM_{10}} * Q_{w,i} * SSA_{w,i}}{\rho_{d,i} * R_{d,i}}}{PM_{10}} \tag{A2}$$

we need to know the dry effective radius ($R_d$) and the dry mass density $\rho_d$, as well as the wet effective radius ($R_w$), the wet extinction efficiency ($Q_w$), and the wet single scattering albedo ($SSA_w$) of a given species. In order to calculate these wet parameters, we need to know the water uptake by a given species at a given RH value. It is therefore necessary to perform hygroscopic calculations for IMPROVE speciation.

We have added "to enable isolation of mass scattering efficiency in our comparisons" to page 13, line 11-12:

"The GEOS-Chem model partitions OM into hydrophilic and hydrophobic fractions, so the same is done for OM measured by IMPROVE to enable isolation of mass scattering efficiency in our comparisons."

*Figures:*

Pg 19:4, Figure 1, Provide the wavelength and RH (or state ambient) for bsp. The sentence starting with "overlaying circles" seems redundant.

These details have been added, and "overlaying circles" has been removed.

Page 22: "Figure 1: Map of IMPROVE sites with collocated scatter ($b_{sp}$) at 550 nm and ambient relative humidity, $PM_{10}$ and $PM_{2.5}$ measurements in North America between 2000-2010."

Pg 20, Figure 2, please include the years included in the comparison, and the wavelength corresponding to the efficiency. It would help to also denote the parts of the figure as well.

The caption of Figure 2 has been changed to include the wavelength of mass scattering efficiency and the years included in the comparison.

Page 23: "Figure 2: Average measured vs calculated $\alpha_{sp}$ at 550 nm at IMPROVE sites between 2000-2010 using GEOS-Chem default optical tables and revised optical tables. The colour of each point corresponds to the average relative humidity at the site. The 1:1 line is black. Slope, offset and correlation coefficient are inset."

Pg 20, Figure 3: Similar comment regarding wavelength as other figures (this is helpful for readers if they are scanning figures and don't have to dig through the text for this information). Please include the years this comparison was made over. Also comment on "default" and "revised".

The caption of Figure 3 has been changed to include the wavelength of mass scattering efficiency and the years included in the comparison. "Default" and "revised" are noted in the caption.

Page 23: "Figure 3: Average measured versus calculated $\alpha_{sp}$ at 550 nm at IMPROVE sites between 2000-2010 using GEOS-Chem default and revised optical tables (Table A1) for measurements taken in 0-35 % RH, 35-65 % RH and 65-95 % RH conditions. The 1:1 line is black. Slope, offset and correlation coefficient are inset."

Pg 21, Figure 4: Include wavelength. Define SIA, OA, PMcoarse. Are these delineations based on modeled or measured composition measurements? Please state in caption. Also include RH ranges (ambient?) and years of comparison.

Wavelength, as well as definitions of SIA, OA and $PM_{coarse}$, and years of comparison have been incorporated into the caption of Figure 4. The delineations are based on IMPROVE composition measurements- this is now stated in the caption.

Mass scattering efficiency is defined for this analysis as $b_{sp}$ (ambient RH) / $PM_{10}$ (40 % RH). It is therefore not appropriate to state an RH range. This definition is used throughout the entire analysis.

Page 24: "Figure 4: Average measured versus calculated $\alpha_{sp}$ (550 nm) at IMPROVE sites between 2000-2010 using GEOS-Chem default and revised optical tables for measurements taken in conditions dominated by secondary inorganic aerosols (SIA), organic aerosols (OA), fine dust, and $PM_{coarse}$ ($PM_{10}$-$PM_{2.5}$). The 1:1 line is black. Slope, offset, and correlation coefficient are inset."

Pg 21, Figure 5: Please include wavelength. What kappa was used for OC and SIA?

The wavelength has been added to the caption. For this figure, water uptake for OA and SIA was calculated using GEOS-Chem default hygroscopic growth factors. To clarify, a sentence has been added.

Page 7 Line 13-14: Water uptake at 80% RH for OA and SIA is calculated using default hygroscopic growth factors from GEOS-Chem.

Pg 22, Figure 6: Please include wavelength and years of comparison.

The caption of Figure 6 has been changed to include the wavelength of mass scattering efficiency and the years included in the comparison.

Page 25: "Figure 6: Average measured versus calculated $a_{sp}$ (550 nm) at IMPROVE sites between 2000-2010 using GEOS-Chem default and revised optical tables for measurements taken in dry conditions (RH<35 %) in winter, spring, summer and fall. The 1:1 line is black. Slope, offset and correlation coefficient are inset."

Pg 23, Figure 8: Please define kappa,s and kappa,o.

$\kappa_o$ and $\kappa_s$ have been defined in the caption of Figure 8.

Page 26: "Figure 8: Slope of measured vs. calculated $\alpha_{sp}$ plot as a function of the $\kappa$ of secondary inorganic aerosols ($\kappa_S$, left) and the $\kappa$ of organic aerosols ($\kappa_O$, right). The line slope=1 is shown in black. $\kappa_s$ and $\kappa_o$ values for which slope=1 are inset."

Pg23, Figure 9: Please define SIA and OA. "Hygroscopic" has a typo.

This typo has been fixed, and SIA and OA have been defined in the caption of figure 9.

Page 26 Line 5: "Hygroscopic growth factor curves for secondary inorganic aerosols (SIA, left) and organic aerosols (OA, right)."

Pg 24, Figure 10: Define SIA and OA.

For clarification, "SIA and OA" has been removed and replaced with "secondary inorganic and organic aerosols" in the caption of Figure 10.

Page 27 "Figure 10: Average relative and absolute change in GEOS-Chem mass scattering efficiency over North America for the year 2006 after implementing revised optical tables for secondary inorganic and organic aerosols."

Pg 25, Figure 11: Please provide wavelength and RH (ambient?), define SIA and OA. This color table makes it hard to see the sites in the Southwest.

The wavelength has been added to the caption, and "SIA and OA" has been replaced with "secondary inorganic and organic aerosols".

Again, RH is ambient for $b_{sp}$, but 40% for $PM_{10}$, so it is not appropriate to provide an RH for mass scattering efficiency.

The colormap has been changed for this plot.

Page 28: "Figure 11: GEOS-Chem annual average mass scattering efficiency (at 550 nm) for the year 2006 using default and revised size and hygroscopicity for secondary inorganic and organic aerosols."

Pg 26: Figure 12: Please provide wavelength and RH (ambient?)

The wavelength has been added to the caption. Again, RH is ambient for $b_{sp}$, but 40% for $PM_{10}$, so it is not appropriate to provide an RH for mass scattering efficiency.

Page 29: "Figure 12: Coincident simulated versus measured average mass scattering efficiency at 550 nm for the year 2006, using default and revised optical tables."

Pg 28, Figure B2: This color table makes it hard to see the comparisons.

The colormap has been changed for this plot.

Pg 30, Table A1, Define SIA and OA, kappa,s and kappa,o (use similar subscripts as SIA and OA?)

SIA, OA, $\kappa_s$, and $\kappa_o$ have been defined in the caption of Table A1.

Page 33 Line 21-23: "Table A1: Default and revised aerosol size and optical properties for secondary inorganic aerosols (SIA) and organic aerosols (OA) at 550 nm at 8 relative humidity values. Columns indicate geometric mean radius ($r_g$), effective radius ($r_{eff}$), extinction efficiency (Q), and single scattering albedo (SSA). $\kappa_s$ and $\kappa_o$ represent the hygroscopic growth parameters for SIA and OA, respectively."

**Reviewer: 2**

The paper studies an important topic for chemistry transport modeling the mass scattering efficiency that affects the conversion from aerosol mass to aerosol extinction and ultimately AOD. Mass scattering efficiency data from IMPROVE is used to constrain a global CTM (GEOS-Chem, GC). It is found that geometric mean radiuses for both dry secondary inorganic and organic aerosols in original GC configuration need to be decreased, and also the aerosol hygroscopicity curve needs to be adjusted to be less hygroscopic. Overall, the paper should be considered for publication after a major revision (and possibly another review).

*General comments*

1) There are lots of averaging done in the data analysis. For example, the captions for Figures 2, 3, 4, and 6 all start with the word 'average', But, how averaging is done is not described in the method section. In addition, how much is the standard deviation in the average?

Clarifications regarding how averaging is done in the analysis have been added and are discussed in detail below. The average mass scattering efficiency at each site is calculated by dividing the average scattering coefficient by the average mass concentration at each site. Variations in these two parameters occur over the sampling period due to variations in relative humidity, composition and size distribution.

2) only R and best fit are shown in all scatter plots. How about RMSE and bias?

RMSE and bias have been added to Figures 2, 3, 4, 6, 12 and B.3.

3) some discussion of uncertainties here are needed. what are the uncertainties in the measurements of scattering coefficient bsp and surface PM2.5? Likewise, do GEOS fields have any systematic bias in simulating RH?

A paragraph discussing uncertainties in PM and $b_{sp}$ have been added to section 2.1.

Page 3 Line 17-22: "The IMPROVE network collects collocated samples at a subset of sites, which can provide insight into precision errors associated with the measurements of major species. Hyslop and White (2008) and Solomon et al. (2014) found mean collocated precision errors ranging from 6-11% for particulate mass measured by IMPROVE. Typical uncertainties in IMPROVE $b_{sp}$ measurements are in the range of 5-15% (Gebhart et al., 2001). Due to nephelometer truncation errors, uncertainties in measured $b_{sp}$ increase as particle size distributions increase, and coarse particle scattering can be underestimated (Molenar, 1997)."

Our analysis is largely independent of GEOS RH since most of our analysis uses IMPROVE RH. We clarified on page 4:

Page 4 Line 8-12: "The majority of our analysis focuses on the accuracy of the GEOS-Chem parameterization of mass scattering efficiency based on optical parameters given in Table A1. These default aerosol physical and optical properties are defined by the Global Aerosol Data Set (GADS) (Koepke et al., 1997), as implemented by Martin et al. (2003), with modifications to dry size distributions (Drury et al., 2010) and dust mass partitioning (Ridley et al., 2012). After evaluating and improving this parameterization, implications are examined using the full GEOS-Chem simulation in section 3.3."

Nonetheless we have added a citation with evaluation of GEOS RH for the interested reader (Gelaro et al., 2017).

*Specific comments*

Pg 4L:10. If IMPROVE PM2.5 is analyzed at 30-50% RH, there will be aerosol water in the aerosol mass measured by IMPROVE. What is the uncertainty here if we assume these are dry particle mass and used in the scattering efficiency calculation? Some consistence is needed or at least discussed between how PM2.5 dry mass is computed in GC vs. how aerosol optics (aerosol hygroscopic growth) are treated in GC. Later in the analysis, the cut off of RH is 35% to be considered as dry (solid) aerosol. Why not 40% as for dry PM2.5?

At 40% RH, we do not assume that $PM_{2.5}$ and $PM_{10}$ mass are dry. The definition of mass scattering efficiency used in our analysis acknowledges this. A line has been added for clarification in section 2.3:

Page 5 Line 11-12: "At 40% RH, hygroscopic components of $PM_{10}$ will have associated water, and thus measured $PM_{10}$ mass is not considered to be dry."

When we simulate mass scattering efficiency in GEOS-Chem in section 3.3, we use the same definition of mass scattering efficiency, with $PM_{10}$ mass at 40% RH in the denominator.

A discussion regarding how aerosol hygroscopic growth is treated in GEOS-Chem can be found on page 9 line 12-18, and GEOS-Chem hygroscopic growth of SIA and OA are shown in Figure 9.

Pg 4:23. Effectively, b_sp is averaged each day, and then daily averages of b_sp is used together with daily PM2.5 to compute scattering efficiency. How annual average of mass scattering efficiency is computed? How the averages are computed for RH in different ranges or for different compositions (e.g., dust dominated, SIA dominated, etc.)? RH does have a strong diurnal variation.

Clarifications regarding how averages are computed overall, for different RH ranges, and for different compositions have been added.

Page 5 Line 16-19: "To reduce the impacts of meteorological variation on the comparison of measured and calculated mass scattering efficiency, we perform averages of hourly $b_{sp,calc}$, $b_{sp,meas}$, and $PM_{10}$ over the entire sampling period at each IMPROVE site $i$. Eq. (3) is then used to obtain average calculated and measured mass scattering efficiency at each site.

$$\alpha_{sp,avg,i} = \frac{b_{sp,avg,i}}{PM_{10,avg,i}} \tag{3}$$"

Page 6, Line 17-19 "The IMPROVE data is divided among the RH groupings using IMPROVE measurements of hourly RH. Within each grouping, average calculated and measured mass scattering efficiencies are obtained for each site using Eq. (3)."

Page 6, Line 24-26: "Using IMPROVE measurements of 24 hr $PM_{2.5}$ mass and speciation and $PM_{10}$ mass, the IMPROVE data is grouped based on dominant aerosol type. Within each group, average calculated and measured mass scattering efficiency is obtained for each site using Eq. (3)."

Pg 6:20-24. The uncertainty of aerosol mass scattering efficiency can also come from the particle composition which essentially affect the particle hygroscopic growth factor, or particle size and density. the scattering efficiency of sulfuric acid vs. ammonium sulfate can be different even for the same size distribution at dry conditions. See "Table 1 in Sensitivity of sulfate direct climate forcing to the hysteresis of particle phase transitions, JGR, 2008". The mass scattering efficiency can also be affected by the mixing state of the particle. The analysis here needs to discuss these uncertainties before focusing on particle size.

We added the following text:

Page 2 Line 4-5: "Mass scattering efficiency is a complex function of aerosol size, composition, hygroscopicity and mixing state (Hand and Malm, 2007; Malm and Kreidenweis, 1997; White, 1986)"

Pg 7:2-3. What is the seasonality of aerosol size distribution?

The seasonality of the aerosol size distribution is described in the previous paragraph, now page 7, line 26-27: "Aerosol surface area and volume distributions fluctuate seasonally in the North Eastern U.S., with summer maxima and winter minima (Stanier et al., 2004)."

Pg 7:13-15. Martin et al's paper showed there can be aerosols in solid phase in RH larger than 40%. The phase transition depends on RH history of the particle, not just RH itself. Could this explain the part of overestimation in GC mass scattering efficiency? What is the fraction of solid SIA particles in U.S.? this paper might be helpful – "global distribution of solid and aqueous sulfate aerosols . . ., JGR, 2008".

The following sentence has been added to acknowledge the possible importance of particle RH history.

Page 11 Line 7: Representation of particle RH history may also be important (Wang et al., 2008).

Pg 8:5. What is r_sp used in figure 8?

The new dry radius of 0.058 μm from section 3.2.2 is used in figure 8. The following sentence has been added to the end of section 3.2.2. to provide clarification:

Page 9 Line 2-3: "For the remainder of the analysis, this new dry radius of 0.058 μm is implemented for SIA and OA."

Pg 11:20. Again, it is the composition that regulates the growth factor and density. The real index of refraction also has the effect on scattering efficiency. In addition, the paper didn't mention effective variance at all. Is it important? References are needed here to support the statements.

A sentence has been revised to note the role of composition.

Page 14 Line 10-12: "In summary, the revised optical properties developed for North America slightly improve the representation of AOD at the global scale, despite the large influence of other factors (e.g. ambient aerosol concentrations and composition) upon AOD."

A sentence has been added on effective variance.

Page 8 Line 19-20: "Effective variance may also be important (Chin et al., 2002) but given sufficient information to simultaneously constrain size and variance, we focus on size.

Pg 13:7-8. The aerosol concentration is the same here for both figures. It is likely that the improvement with new optics is within the range of inter-annual variability of alpha_sp itself? It will be more convincing that the validation is done using other years data (say 2007 or 2008).

To repeat the analysis for additional years would be a substantive undertaking. This has been added as a suggestion for future work (page 12, line 3).

Figure 5. Caption. Is aerosol effective radius shown for dry (solid) or wet particles?

Aerosol effective radius is shown for wet particles, clarification has been added to the caption of Figure 5:

Page 24: "Figure 5: Mass scattering efficiency ($\alpha_{sp}$) at 550 nm as a function of aerosol wet effective radius for organic aerosol and secondary inorganic aerosol."

Figure 9. The GADS assume sulfate as 75% H2SO4, which often has the largest hygroscopic growth. For lab data, please provide reference. Also, lab data clearly shows that ammonium sulfate particles can be in solid phase in range of RH between 40-80%, which may explain why GC has an overestimation in scattering efficiency. How large is the uncertainty in assuming that all SIA are aqueous in RH of 40-80%? If particles are in solid phase in upper troposphere (as models suggest), the large improvement in surface with new optics may not be reflected in the AOD comparison.

The reference has been added for the lab data in the caption of Figure 9.

To address the possible importance of RH history, the following sentence has been added.

Page 11 Line 7: "Representation of particle RH history may also be important (Wang et al., 2008)"

Figure 12. add RMSE, and also, how the spectral AOD (or Angstrom exponent) is compared against AERONET? This can be an interesting test for new optics that is a result of adjusting particle size. Particle size affects both scattering efficiency as well as the spectral AOD slope. Ideally, the new optics should provide an overall improvements for the model.

RMSE and bias have been added. Aeronet spectral AOD, or Angstrom exponent, largely reflect the ratio of coarse to fine mass and is beyond the scope of this analysis.

[revised manuscript text omitted]

**Page 39: [2] Deleted**      **Robyn Latimer**      **10/24/18 4:29:00 PM**

[Figure]

**Page 40: [3] Deleted**      **Robyn Latimer**      **10/24/18 4:29:00 PM**

[Figure]

Page 43: [4] Deleted        Robyn Latimer        10/28/18 7:06:00 PM

$\alpha_{sp}$ relative change (%), 2006

$\alpha_{sp}$ absolute change (m$^2$ g$^{-1}$), 2006

Page 46: [5] Deleted        Robyn Latimer        10/28/18 7:07:00 PM